# PANOPTIC PAIRWISE DISTORTION GRAPH

**Muhammad Kamran Janjua, Abdul Wahab, Bahador Rashidi**
Huawei Technologies, Canada
{muhammad.kamran.janjua}@huawei.com
🐼 aismartperception.github.io/distortion-graph/

## ABSTRACT

In this work, we introduce a new perspective on comparative image assessment by representing an image pair as a structured composition of its regions. In contrast, existing methods focus on whole image analysis, while implicitly relying on region-level understanding. We extend the intra-image notion of a scene graph to inter-image, and propose a novel task of Distortion Graph (DG). DG treats paired images as a structured topology grounded in regions, and represents dense degradation information such as distortion type, severity, comparison and quality score in a compact interpretable graph structure. To realize the task of learning a distortion graph, we contribute (i) a region-level dataset, PANDASET, (ii) a benchmark suite, PANDABENCH, with varying region-level difficulty, and (iii) an efficient architecture, PANDA, to generate distortion graphs. We demonstrate that PANDABENCH poses a significant challenge for state-of-the-art multimodal large language models (MLLMs) as they fail to understand region-level degradations even when fed with explicit region cues. We show that training on PANDASET or prompting with DG elicits region-wise distortion understanding, opening a new direction for fine-grained, structured pairwise image assessment.

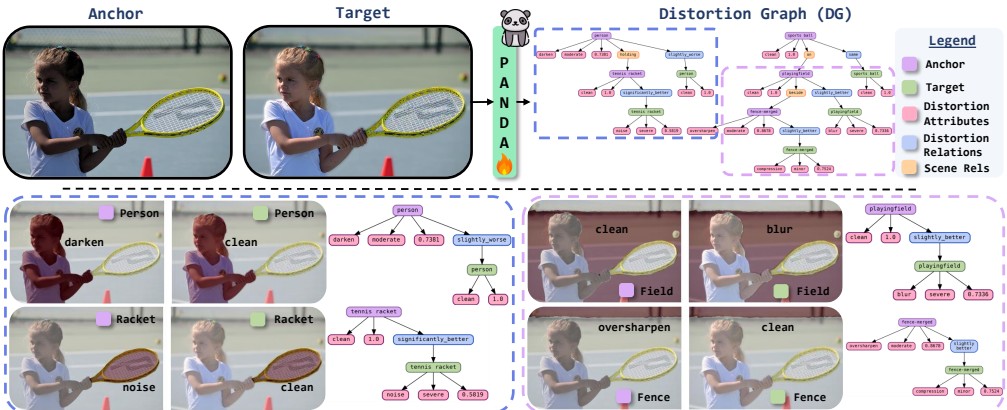

Figure 1: **DG Task Overview.** Top: Given two images, PANDA learns the proposed Distortion Graph (DG). Bottom: Grounded Subgraphs illustrate how DG grounds regions in terms of distortion information.

## 1 INTRODUCTION

In humans, perceptual decisions[1] are often cognitively involved, deliberate, and contextual (Ding & Gold, 2013). Studies have argued that any model of such perceptual decision making should consider the representation of the relevant sensory input and how that representation is formed (Gold & Ding, 2013). In the case of visual stimuli, one example of such perceptual decisions is distortion analysis. Yet, when it comes to computational perceptual decision making, often, current design choices favor a top-down approach considering a global view of the input for analyses tasks (Zhang et al., 2025; Li et al., 2025; Wu et al., 2024c; You et al., 2024b; Wu et al., 2023).

---

[1] any categorical decision about the presence or identity of a sensory stimulus

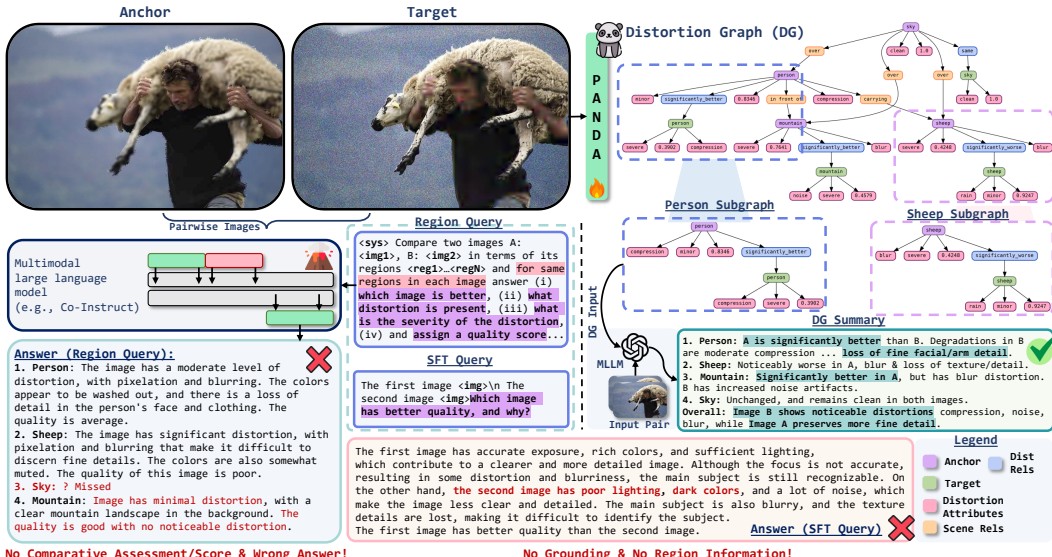

Figure 2: **Motivation.** Current MLLMs (e.g., Co-Instruct (2024c)) fail at region-level understanding, struggling even when given explicit region details (name, description, bounding box). DG grounds assessment in regions, relating distortions and attributes to provide a structured view. Optionally, the graph can be fed to an MLLM for region-wise language descriptions. Scene relations (yellow) are not predicted. Best viewed zoomed-in.

Such design choices are inherently limiting because they do not lend themselves to fine-grained understanding (Rahmanzadehgervi et al., 2024). Further, in multimodal language models (MLLMs), they restrict distortion-specific visual question answering (VQA), ranking, descriptive understanding, and even quality scoring to image-level (Li et al., 2025; Jiang et al., 2024; You et al., 2024b). With instruction tuning (Liu et al., 2023) on a limited instruction set as the learning paradigm, more often than not, the outcome is a rigid multi-billion parameter model that parrots template responses (Zeng et al., 2023; Chu et al., 2025), see fig. 2. One reason for the prevalence of a top-down approach is the lack of a structured representation that is grounded in image regions.

In this work, we offer a novel perspective on learning a structured representation between image pairs for assessment, and introduce the *task of a Distortion Graph (*DG*)*. DG is a general-purpose pairwise graph structure, with regions as atomic components. Each node corresponds to a region, while inter-region edges capture comparative relationships (predicates). Nodes encode the local distortion and severity type as well as a region-level quality score (attributes), enabling region-first reasoning over paired images, see fig. 1. We argue that DG is a pertinent structured approach for pairwise comparative purposes since such information aggregates to image-level judgments, while vice-versa is often not true. We position DG such that it can complement MLLMs in offering region-wise distortion analysis in natural language, see fig. 2 for illustration.

To realize the task of learning a distortion graph, we introduce a region-level distortion dataset, termed as PANDASET. By design, PANDASET comprises over 500K image pairs degraded by 15 different distortions, ranging from sensor-induced and equipment failure distortions to weather distortions, with four different severity levels. Each region has an associated quality score indicating on what end of the distortion spectrum (from clean to severely degraded) it lies. We show that the proposed task is indeed computationally tractable and design an efficient architecture, termed PANDA, that learns to predict region-level attributes and predicates to generate DG.

We introduce PANDABENCH, a benchmark derived from PANDASET with three splits of increasing region-level difficulty to enable systematic evaluation. We evaluate both open-source and closed-source frontier MLLMs on the proposed benchmark under zero-shot and fine-tuned setups. We empirically show that distortion-specific MLLMs suffer greatly when reasoning over image regions and often resort to template responses, even when explicitly prompted with region-wise visual markers. Further, such methods are limited by the context length in terms of processing variable number of regions. On the other hand, frontier MLLMs are less rigid and have superior instruction fol-

lowing abilities, yet their performance is not much better than random chance. Additionally, as a showcase application, we demonstrate that DG in chain-of-thought prompting encourages emergent capabilities of MLLMs for distortion understanding.

## 2 RELATED WORK

**Distortion MLLMs.** One of the earliest efforts towards enabling low-level vision understanding in MLLMs is Q-Instruct, wherein Wu et al. (2024b) introduced a new dataset, Q-Pathways, and instruction-tuned LLaVA-v1.5 (Liu et al., 2024a) for distortion identification and VQA. Several works followed suit, introducing improved benchmarks, training recipes, and methods. Zhang et al. (2025) proposed an extension to Q-Pathways, and unified the task of image quality assessment in terms of numerical scores and descriptive analysis in MLLMs. Wu et al. (2024a) introduced a benchmark Q-Bench comprising low-level attribute and descriptive tasks, along with quality score regression to evaluate MLLMs' ability on low-level vision. However, all of these works focus on single image analysis. Towards comparative assessment, You et al. (2024b) proposed a new dataset, M-BAPPS, and fine-tuned Vicuna-v1.5 (Chiang et al., 2023) on quality comparison task in the full-reference setting (i.e., a clean reference image should be available). DepictQA (You et al., 2024a) introduced a general-purpose dataset, DQ495K, to let MLLMs perform similar comparative tasks but even without any reference image. Co-Instruct (Wu et al., 2024c) introduced MICBench, a benchmark to evaluate MLLMs performance on comparative tasks, but the introduced method allowed multiple images to be compared. Note that specialist models in pre-MLLM era often focused on learning a numerical score for image quality assessment (IQA), and several works exist in literature along this direction (Chen et al., 2024a; Agnolucci et al., 2024; Wang et al., 2023a).

**Region Understanding.** None of the above mentioned work is region-first by design. Further, the MLLMs fine-tuned on these datasets can not extend to regions since that is an out-of-domain setting (Rajani et al., 2025), also see fig. 2. Literature, however, has made efforts to enable general-purpose region-level understanding in MLLMs. Set-of-Mark (SoM) prompting (Yang et al., 2023) introduced visual markers overlaid on regions to prompt MLLMs to reason about regions. Wang et al. (2023b) proposed to utilize region-of-interest (RoI) features to generate region-level tokens for MLLMs. Omni-RGPT (Heo et al., 2025) introduced token markers for region-level comprehension in images and videos. One particular work, Seagull (Chen et al., 2024c), explored region-level descriptive distortion analysis, but only for single image setting. Seagull utilized mask pooling to generate region-level tokens for MLLMs, and introduced a dataset for the analysis task. Q-Ground (Chen et al., 2024b) introduced QGround100K, built on top of Q-Instruct (Wu et al., 2023), a single-image dataset of image, textual descriptions, and region-level segmentation and trained an MLLM to jointly provide the explanation, and pixel-level distortion masks for 5 distortion types. Its grounding is thus phrased as mapping quality descriptions onto segmentation masks within one image. Similarly, Grounding-IQA (Chen et al., 2024d) operates in a single-image setting, and defined two sub-tasks: GIQA-DES which considers quality descriptions with bounding boxes and GIQA-VQA which refers to region-wise quality QA. It introduced a dataset GIQA-160K plus GIQA-Bench to fine-tune and evaluate MLLMs on grounding quality attributes to local regions.

Note that none of these works are simultaneously (i) comparative in nature, (ii) region-first, and (iii) provide dense distortion annotations, for a diverse set of distortions, at the region level (distortion type, severity, quality scores) plus region-wise comparative labels between two images.

## 3 DISTORTION GRAPH

Consider a pair of images denoted by $\mathbf{I}_A$ and $\mathbf{I}_T$ referred to as anchor and target, respectively. A Distortion Graph (DG) is defined as a 4-tuple, i.e.,

$$G = (\mathbb{O}^{\mathbf{I}_A}, \mathbb{O}^{\mathbf{I}_T}, \mathbb{E}_D, \mathbb{E}_S), \tag{1}$$

where $\mathbb{O}^{\mathbf{I}_A}, \mathbb{O}^{\mathbf{I}_T}$ are sets of object (or regions) nodes in images $\mathbf{I}_A$ (anchor), and $\mathbf{I}_T$ (target), respectively. $\mathbb{E}_D$ and $\mathbb{E}_S$ are sets of distortion and scene edges denoting relations among the objects. Given a set of distortion relations denoted by $\mathbb{R}_D$, we can formally say $\mathbb{E}_D \subseteq \mathbb{O}^{\mathbf{I}_A} \times \mathbb{R}_D \times \mathbb{O}^{\mathbf{I}_T}$. Similarly, given a set of scene relations denoted by $\mathbb{R}_S$, we can write $\mathbb{E}_S \subseteq (\mathbb{O}^{\mathbf{I}_A} \times \mathbb{R}_S \times \mathbb{O}^{\mathbf{I}_A}) \cup (\mathbb{O}^{\mathbf{I}_T} \times \mathbb{R}_S \times \mathbb{O}^{\mathbf{I}_T})$, where $\mathbb{O} := \mathbb{O}^{\mathbf{I}_A} \cup \mathbb{O}^{\mathbf{I}_T}$.

Let $\mathbb{A}_D$ denote the set of distortion attributes, and $\mathbb{A}_S$ denote the set of scene attributes, then each object $o_i^j \in \mathbb{O}$ takes the form $o_i^j = (c_i^j, m_i^j, \mathbf{I}_j, \mathbb{A}_{D,i}, \mathbb{A}_{S,i})$, where $j \in \{A, T\}$, $c_i^j$ is the class of the object, $\mathbf{I}_j$ denotes the image the object belongs to, $\mathbb{A}_{D,i} \subseteq \mathbb{A}_D$, and $\mathbb{A}_{S,i} \subseteq \mathbb{A}_S$. Let $\gamma$ denote a map written as $\gamma : \mathbb{O} \to \mathbb{M}$ with $\mathbb{M}$ denoting a set of binary masks and $\mathbb{M} := \mathbb{M}^{\mathbf{I}_A} \cup \mathbb{M}^{\mathbf{I}_T}$. In other words, $\gamma$ maps each object $o_i^j \in \mathbb{O}$ to its binary mask $m_i^j \in \mathbb{M}$, i.e., $\gamma(o_i^j) = m_i^j$, effectively grounding the object in its image. Note that, in eq. (1), $\mathbb{E}_S$ is optional, and subsequently, $\mathbb{R}_S$ and $\mathbb{A}_S$ are also optional, since distortion graph generalizes scene graph (Johnson et al., 2015; Li et al., 2024) to distortions and scene information (relations or attributes) is orthogonal to DG semantics. For the sake of completeness, however, we define DG with scene information.

## 3.1 PROPERTIES OF DISTORTION GRAPH

A Distortion Graph (DG) obeys three important properties to meaningfully describe an image pair in terms of its regions, namely the validity, ordering, and functional comparison.

**Preliminaries.** There exists a finite index set $\mathbb{J}$ and two injective enumerations $(o_i^A)_{i \in \mathbb{J}} \subseteq \mathbb{O}^{\mathbf{I}_A}$ and $(o_i^T)_{i \in \mathbb{J}} \subseteq \mathbb{O}^{\mathbf{I}_T}$, such that for each $i \in \mathbb{J}$, $o_i^A$ and $o_i^T$ denote the same object (or region) across two images (anchor and target). By convention, if $i \neq j$ then $o_i^A \neq o_j^A$ and $o_i^T \neq o_j^T$. We refer to $(o_i^A, o_i^T)$ as the $i$th matched region (or object) pair.

**Definition 1** (Validity of Distortion Edges). For every $(o, r, o') \in \mathbb{E}_D$, there exists $i \in \mathbb{J}$ with $o = o_i^A \in \mathbb{O}^{\mathbf{I}_A}$, $o' = o_i^T \in \mathbb{O}^{\mathbf{I}_T}$, and $r \in \mathbb{R}_D$. In particular, no intra-image triplets belong to $\mathbb{E}_D$. Formally, we define validity as:

$$\mathbb{E}_D \subseteq \{(o_i^A, r, o_i^T) \ : \ i \in \mathbb{J}, r \in \mathbb{R}_D\} \subseteq \mathbb{O}^{\mathbf{I}_A} \times \mathbb{R}_D \times \mathbb{O}^{\mathbf{I}_T}. \tag{2}$$

**Definition 2** (Ordering of Distortion Relations). The distortion relation set (or comparative relation set) $\mathbb{R}_D$ is interpreted as *anchor relative to target*. Accordingly, distortion edges are always ordered and written as $(o_i^A, r, o_i^T)$. Formally,

$$\mathbb{E}_D \subseteq \mathbb{O}^{\mathbf{I}_A} \times \mathbb{R}_D \times \mathbb{O}^{\mathbf{I}_T} \quad \text{and} \quad \forall i \in \mathbb{J}, \forall r \in \mathbb{R}_D \ : \ (o_i^T, r, o_i^A) \notin \mathbb{E}_D. \tag{3}$$

**Definition 3** (Functional Comparison). For every matched region pair $(o_i^A, o_i^T)$ where $i \in \mathbb{J}$, exactly one distortion relation $r \in \mathbb{R}_D$ compares them. Formally, we can write:

$$\forall i \in \mathbb{J} \ : \ \left| \{r \in \mathbb{R}_D \ : \ (o_i^A, r, o_i^T) \in \mathbb{E}_D\} \right| = 1. \tag{4}$$

## 3.2 GENERATING DISTORTION GRAPH

We propose a simple and efficient method, termed as PANDA to learn **Pan**optic Pairwise **D**istortion Gr**a**ph for an image pair. PANDA is a neural network parametrized by $\theta$ that takes input a pair of images, referred to as anchor and target, and predicts for each region, distortion relationship (comparative relation), type of distortion afflicting the region, severity of the distortion, and a quality score through multiple heads in DETR-like (Carion et al., 2020) fashion, see fig. 3 for illustration of the architecture.

We treat comparative relation, distortion and severity type as categorical values with categorical cross-entropy as the loss function of choice for their respective heads, while $L_1$ loss function penalizes the score regression head. Each head is a simple 3-layer MLP. PANDA is trained for a total of 30 epochs with AdamW (Loshchilov & Hutter, 2017) as the optimizer, and a learning rate of $1e-4$ with weight decay of 0.01. The total loss function is $\mathcal{L} = \lambda_1 L_{CE}^{\text{rel}} + \lambda_2 L_{CE}^{\text{dist}} + \lambda_3 L_{CE}^{\text{sev}} + \lambda_4 L_1^{\text{score}}$. We search for the optimal values of the learning rate, and each $\lambda$; see details in appendix E.

### 3.2.1 PANDA ARCHITECTURE

Given an image pair $\mathbf{I}_A$ and $\mathbf{I}_T$, we feed both to a pretrained encoder (e.g., DINOv2 (Oquab et al., 2023)) to get a feature map $\mathbf{F}^j \in \mathcal{R}^{H \times W \times C}$, where $H, W$ denote the spatial dimensions, $C$ is the number of channels, and $j \in \{\mathbf{I}_A, \mathbf{I}_T\}$. A panoptic segmentation method (e.g., SAM (Kirillov et al., 2023)) acts as the map function ($\gamma$) to segment each region into corresponding binary masks $m_i^j \in \mathbb{M}$. Let $N_R$ denotes the number of regions in each image. We make sure that all regions align

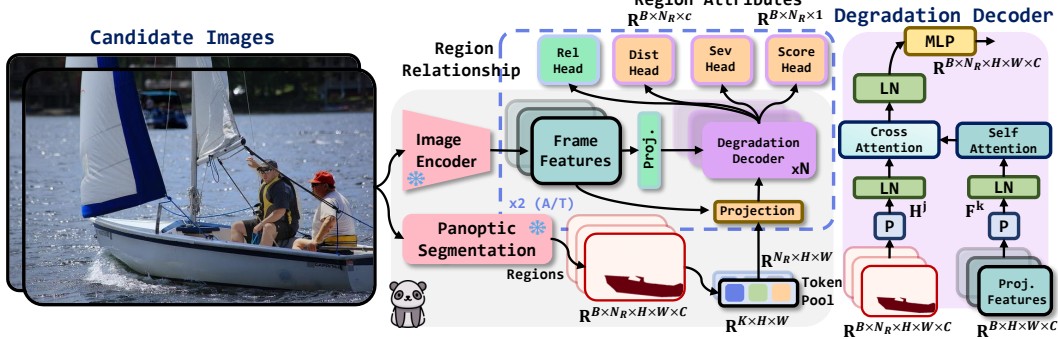

Figure 3: **Architecture Diagram.** Illustration of the proposed PANDA architecture to learn Distortion Graph (DG). A pair of image is fed as input, and for each region in the pair, their comparative relationship (predicates), distortion type, severity type and quality score (attributes) are predicted.

across both images in the pair for one-to-one correspondence in regions, i.e., $N_R = N_R^A = N_R^T$. For exposition, we only use $N_R$ here onward.

**Token Pool.** To associate each region with the image, we maintain a token pool comprising learnable vectors called tokens of same spatial shape as the binary masks, similar in spirit to Heo et al. (2025). We define a token pool as a set of learnable vectors for each image in the pair $\mathbb{T}^{\mathbf{I}_A}$ and $\mathbb{T}^{\mathbf{I}_T}$ wherein each $t_i^j \in \mathcal{R}^{H \times W \times 1}$, $j \in \{\mathbf{I}_A, \mathbf{I}_T\}$, and $\left|\mathbb{T}^{\mathbf{I}_A}\right| = \left|\mathbb{T}^{\mathbf{I}_T}\right| = K$ where $K$ is the total number of tokens. From their respective pools, we sample $N_R$ indices uniformly and without replacement to obtain $\mathbb{T}_{N_R}^{\mathbf{I}_A} = \{t_i^{\mathbf{I}_A}\}_{i=1}^{N_R}$ and $\mathbb{T}_{N_R}^{\mathbf{I}_T} = \{t_i^{\mathbf{I}_T}\}_{i=1}^{N_R}$ or more generally with abuse of notation, we can write $t_i^j \in \mathbb{T}_{N_R}^j \in \mathcal{R}^{N_R \times H \times W}$. Every $i$th region $m_i^j$ is one-to-one matched with $i$th token $t_i^j$. We then compute the Hadamard product $h_i^j = m_i^j \odot t_i^j$ where $h_i^j \in \mathbf{H}^j$ and $\mathbf{H}^j \in \mathcal{R}^{N_R \times H \times W}$, and project it with a convolutional layer to match the dimensions and combine it with the respective image features, i.e., $\hat{\mathbf{H}}^j = \texttt{Conv}(\mathbf{H}^j) \odot \mathbf{F}^j \in \mathcal{R}^{N_R \times H \times W \times C}$.

Additionally, we let the pretrained features be learnable through a $1 \times 1$ convolutional layer and obtain $\hat{\mathbf{F}}^j = \texttt{Conv}(\mathbf{F}^j)$. This procedure allows variable number of regions to borrow information from respective images with minimal compute.

**Degradation Decoder.** Given the feature map for a batch $B$ of image pairs $\hat{\mathbf{F}}^k$ where $k \in \{\mathbf{I}_A, \mathbf{I}_T\}$, and the region features $\hat{\mathbf{H}}^j$ where $j \in \{\mathbf{I}_A, \mathbf{I}_T\}$ and $j \neq k$, we feed them through $L$ Transformer layers (Dosovitskiy et al., 2020) followed by four prediction heads to decode each region into relations and distortion attributes. At $l$-th layer, we first reshape $\hat{\mathbf{F}}^k \in \mathcal{R}^{B \times H \times W \times C}$ to $\hat{\mathbf{F}}^k \in \mathcal{R}^{B \times D \times C}$ where $D = H \times W$ and denotes the number of patches, and add positional embedding to each patch, i.e., $\hat{\mathbf{F}}^k = \hat{\mathbf{F}}^k + \texttt{PE}$. We, then, project $\hat{\mathbf{F}}^k$ to obtain $\mathbf{Q}_{\hat{\mathbf{F}}^k}, \mathbf{K}_{\hat{\mathbf{F}}^k}, \mathbf{V}_{\hat{\mathbf{F}}^k} \leftarrow \hat{\mathbf{F}}^k W^{\hat{\mathbf{F}}^k}$ matrices, and compute multi-head attention (MHA) (Vaswani et al., 2017) followed by a skip connection to obtain

$$\mathbf{y}_{\hat{\mathbf{F}}^k}^{\text{SA}} = \left[\texttt{MHA}(\mathbf{Q}_{\hat{\mathbf{F}}^k}, \mathbf{K}_{\hat{\mathbf{F}}^k}, \mathbf{V}_{\hat{\mathbf{F}}^k}) + \hat{\mathbf{F}}^k\right] \in \mathcal{R}^{B \times D \times C}. \tag{5}$$

We then let each region in image $j$ attend to the image features and learn its correspondence with its matched region in the other image $k$. In other words, we compute cross-attention where query comes from $\hat{\mathbf{H}}^j$, and key and value matrices come from $\hat{\mathbf{F}}^k$. Similar to $\hat{\mathbf{F}}^k$, we reshape $\hat{\mathbf{H}}^j \in \mathcal{R}^{B \times N_R \times H \times W \times C}$ to $\mathcal{R}^{(B \times N_R) \times D \times C}$ by combining regions in the batch dimension and $D$ denotes the number of patches of each region. For each region in batch dimension and for each patch associated with the region, we add positional embedding, i.e., $\hat{\mathbf{H}}^j = \hat{\mathbf{H}}^j + \texttt{PE}_{N_R} + \texttt{PE}$. We, then, project $\hat{\mathbf{H}}^j$ to obtain the query matrix $\mathbf{Q}_{\hat{\mathbf{H}}^j} \leftarrow \hat{\mathbf{H}}^j W^{\hat{\mathbf{H}}^j}$, and key and value matrices come from $\mathbf{y}_{\hat{\mathbf{F}}^k}^{\text{SA}}$, i.e., $\mathbf{K}_{\mathbf{y}^{\text{SA}}}, \mathbf{V}_{\mathbf{y}^{\text{SA}}} \leftarrow \mathbf{y}^{\text{SA}} W^{\mathbf{y}^{\text{SA}}}$. For brevity, we drop the subscript $\hat{\mathbf{F}}^k$ from $\mathbf{y}^{\text{SA}}$. Since $Q_{\hat{\mathbf{H}}^j} \in \mathcal{R}^{(B \times N_R) \times D \times C}$, we repeat both $\mathbf{K}_{\mathbf{y}^{\text{SA}}}$ and $\mathbf{V}_{\mathbf{y}^{\text{SA}}}$ $N_R$ times and compute multi-head cross-attention followed by a skip connection, i.e.,

$$\mathbf{y}_{j \to k}^{\text{CA}} = \left[\texttt{MHA}(\mathbf{Q}_{\hat{\mathbf{H}}^j}, \mathbf{K}_{\mathbf{y}^{\text{SA}}}, \mathbf{V}_{\mathbf{y}^{\text{SA}}}) + \hat{\mathbf{H}}^j\right] \in \mathcal{R}^{(B \times N_R) \times D \times C}. \tag{6}$$

| Benchmark | Region First | Comparative Nature | Diverse Distortions | Severity Levels | Quality Score |
|---|:---:|:---:|:---:|:---:|:---:|
| Q-Bench (Wu et al., 2024a) | ✗ | ✗ | ✗ | ✗ | ✓ |
| DQ495K (You et al., 2024a) | ✗ | ✓ | ✓ | ✓ | ✗ |
| Seagull-100w (Chen et al., 2024c) | ✓ | ✗ | ✗ | ✓ | ✓ |
| Q-Pathways (Wu et al., 2023) | ✗ | ✗ | ✓ | ✗ | ✓ |
| MICBench (Wu et al., 2024c) | ✗ | ✓ | ✓ | ✗ | ✓ |
| Q-Ground100K (Chen et al., 2024b) | ✓ | ✗ | ✗ | ✗ | ✓ |
| GIQA-Bench (Chen et al., 2024d) | ✓ | ✗ | ✗ | ✗ | ✗ |
| 🐼 PANDABENCH | ✓ | ✓ | ✓ | ✓ | ✓ |

Table 1: **Benchmark Summary.** A comparison of PANDABENCH with prior distortion benchmarks in literature. Note that, none of these benchmarks are both region-first and comparative by design.

The output feature map goes through an MLP and we obtain $\mathbf{y}_{j\to k} = \texttt{MLP}(\mathbf{y}_{j\to k}^{\text{CA}})$ which summarizes how each region in image $j$ compares with image $k$. In other words, such a procedure lets each region in one image find its corresponding region in the other image in a pair. Note that, before attention and MLP layers, we project the input through a layernorm (Ba et al., 2016).[2]

**Prediction Heads.** A simple global average pool (GAP) averages the spatial dimension of the obtained output feature map, i.e., $\mathbf{G}_{j\to k} = \texttt{GAP}(\mathbf{y}_{j\to k}) \in \mathcal{R}^{B\times N_R \times C}$. We feed $\mathbf{G}_{j\to k}$ to four 3-layer MLPs with layernorm (Ba et al., 2016) and GELU activation (Hendrycks & Gimpel, 2016), i.e., $\mathbf{y} = \texttt{GELU}(\texttt{LN}(\texttt{FC}(\mathbf{G}_{j\to k})))$ followed by $\hat{\mathbf{y}} = \texttt{FC}(\texttt{GELU}(\texttt{LN}(\texttt{FC}(\mathbf{y})))) \in \mathcal{R}^{B\times N_R \times c}$, where LN is layernorm, FC is a fully-connected layer, $\hat{\mathbf{y}}$ is the output, and $c$ is the output dimension which can either be the number of classes or a single score accordingly. We omit scene prediction heads, as scene information is out of scope, though the architecture trivially accommodates them.

## 4 DATASET & BENCHMARK

Given the lack of a dataset for the purpose of region-level pairwise comparative distortion analysis, see table 1, we propose a new dataset, termed as PANDASET, and a benchmark, termed as PANDABENCH. We build our dataset on two publicly available datasets, namely PSG (Yang et al., 2022), and Seagull-100w (Chen et al., 2024c). PSG (Yang et al., 2022) is an intersection of Visual Genome (Krishna et al., 2017) with COCO (Lin et al., 2014), i.e., combining scene information with region-level panoptic segmentation. While Seagull-100w contains images with real distortions, simulated through varying the parameters of an ISP, and region-wise segmentation maps.

### 4.1 PANDASET

We sample $2,200$ high-quality unique images depicting diverse set of scenes in both indoor and outdoor settings captured in various lighting settings with different camera angles. Around $1,592$ images are taken from PSG, and $608$ images come from Seagull-100w. We divide the dataset into train, validation and test sets with $2,000$ images in train, $50$ images in validation, and $150$ images in test set. Each image has variable number of regions with a maximum of $112$, and a mean of $18$. In total, PANDASET contains $528$K image pairs across train, validation and test sets.

**Distortions.** We extend 11 categories of distortions from DepictQA (You et al., 2024a) with three weather-induced distortions, namely, rain, snow and haze, yielding a total of 14 distortion categories: *blur, brightness, compression, contrast strengthen, contrast weaken, darken, haze, noise, oversharpen, pixelate, rain, saturation strengthen, saturation weaken, and snow*. Each distortion is further sub-categorized (different types of noise, blur, compression methods, etc.), giving a total of 32 sub-types. We also consider the mixed distortion setting, where each region is degraded differently by uniformly sampling from the list of distortions. In case of Seagull-100w, however, we keep the ISP degradation wherever an overlap exists with the chosen distortion for a particular region, i.e., ISP noise or blur for a region is picked over synthetic noise or blur.

---

[2]We use pre-norm residual blocks.

| Methods | Comparison | | | | Distortion | | | | Severity | | | | Scores | |
|---|---|---|---|---|---|---|---|---|---|---|---|---|---|---|
| | A | P | R | F1 | A | P | R | F1 | A | P | R | F1 | SR | PL |
| 🖻 Q-SiT (2025) | – | – | – | – | 0.18 | 0.09 | 0.07 | 0.05 | – | – | – | – | – | – |
| 🖻 Q-Insight (2025) | – | – | – | – | 0.16 | 0.16 | 0.09 | 0.07 | 0.19 | 0.19 | 0.24 | 0.11 | – | – |
| 🖻 DepictQA (2024b) | – | – | – | – | 0.15 | 0.11 | 0.13 | 0.10 | 0.27 | 0.12 | 0.20 | 0.11 | – | – |
| 🖻 Seagull (2024c) | – | – | – | – | 0.23 | 0.25 | 0.20 | 0.18 | 0.32 | 0.34 | 0.26 | 0.26 | – | – |
| 🖻 Gemma-3 27B (2025) | – | – | – | – | 0.27 | 0.31 | 0.24 | 0.20 | 0.30 | 0.31 | 0.30 | 0.27 | – | – |
| 🖻 DepictQA† (2024b) | 0.49 | 0.48 | 0.38 | 0.42 | 0.75 | 0.82 | 0.71 | 0.76 | 0.55 | 0.53 | 0.45 | 0.48 | 0.78 | 0.77 |
| 🔒 GPT-5 Nano (2025) | 0.34 | 0.26 | 0.28 | 0.26 | 0.37 | 0.37 | 0.29 | 0.28 | 0.29 | 0.28 | 0.27 | 0.21 | 0.39 | 0.44 |
| 🔒 GPT-5 Mini (2025) | 0.31 | 0.32 | 0.31 | 0.26 | 0.49 | 0.54 | 0.44 | 0.44 | 0.36 | 0.32 | 0.31 | 0.29 | 0.52 | 0.54 |
| 🔒 GPT-4o (2024) | 0.26 | 0.29 | 0.26 | 0.23 | 0.46 | 0.60 | 0.41 | 0.44 | 0.33 | 0.34 | 0.29 | 0.27 | 0.54 | 0.56 |
| 🔒 Gemini 2.5 Pro (2025) | 0.22 | 0.29 | 0.25 | 0.18 | 0.39 | 0.59 | 0.36 | 0.41 | 0.29 | 0.32 | 0.25 | 0.26 | 0.59 | 0.60 |
| 📥 Random | 0.20 | 0.20 | 0.20 | 0.19 | 0.07 | 0.07 | 0.07 | 0.06 | 0.25 | 0.25 | 0.25 | 0.25 | 0.00 | 0.00 |
| 📥 Linear Probe | 0.37 | 0.35 | 0.22 | 0.15 | 0.20 | 0.16 | 0.09 | 0.07 | 0.27 | 0.25 | 0.26 | 0.15 | 0.12 | 0.14 |
| 📥 Attentive Probe | 0.47 | 0.47 | 0.42 | 0.43 | 0.40 | 0.38 | 0.42 | 0.39 | 0.29 | 0.26 | 0.27 | 0.26 | 0.37 | 0.44 |
| 🐼 **PANDA** | **0.58** | **0.61** | **0.54** | **0.56** | **0.78** | **0.79** | **0.81** | **0.79** | **0.59** | **0.61** | **0.58** | **0.59** | **0.79** | **0.83** |

Table 2: **PANDABENCH Easy.** Results of different MLLMs on the Easy set. 🖻 indicates open source/open-weight, and 🔒 denotes closed-source MLLMs, 📥 stands for baselines. † indicates method is trained on PANDASET. **A**: Accuracy, **P**: Precision, **R**: Recall, **SR**: SRCC & **PL**: PLCC.

This forms the basis of pairs wherein we sample two images with different distortions but same scenes forming a total of $^{16}P_2 = 240$ permutations, and, hence, 480K pairs for training, 12K pairs for validation, and 36K pairs for testing. We add distortions region-wise by uniformly sampling from 14 distortions with $80\%$ probability that a region is degraded, and $20\%$ probability that it is clean. A region can either be degraded with one of the 14 distortions or it can be clean giving a total of 15 different distortion types; samples are shown in fig. 8.

**Severity & Quality Scores.** For each region, the chosen distortion is added with one of three severity levels: *minor, moderate, and severe*. In case of no distortion, clean with $20\%$ probability, the severity is set to none, giving a total of four severity types. The intensity of each distortion varies with each severity level, and we follow You et al. (2024a) to vary the intensity of non-weather distortions. For weather-induced distortions such as rain, and snow, we utilize various rain and snow overlays (Garg & Nayar, 2006; Liu et al., 2018), while for haze, we vary the atmospheric light and haze density parameters in the atmospheric scattering model following Guo et al. (2024). For quality scores, we compute full-reference TOPIQ (Chen et al., 2024a) score ($\in [0, 1]$) between the distorted region and the ground-truth region to serve as a quantitative indication of region quality. We present a visual summary of the entire PANDASET in fig. 6 (appendix). Regions are uniformly distributed among different distortions, around $\approx 3.5\%$, and each severity category spans $\approx 15\%$ of regions.

**Comparative Relationships.** In DG, inter-region edges are labeled with relationships (or predicates) that compare them. We find that TOPIQ (Chen et al., 2024a) accurately indicates the severity of a distortion in terms of a numerical score. For simplicity we adopt TOPIQ as the basis of comparative relationships, however, more complex preferences can be used. For every region pair, we define a threshold on the difference between scores of the region in the anchor and the target image. If the difference is less than $|0.1|$, we label the region as *same*, while it is *slightly better or worse* in the interval $\pm[0.1, 0.3)$. Similarly, if the difference is more than $0.3$, we label it as *significantly better or worse* depending on which pair (anchor or target) scores higher.

## 4.2 PANDABENCH

Three representative splits from the test set of PANDASET, termed Easy, Medium, and Hard, comprise the proposed benchmark, PANDABENCH. In Easy, we only consider pairs where all the regions in an image are degraded by a single type of distortion, but with either same or different severity levels. In Medium, one of the images in the pair is from the mixed setting, i.e., each region exhibits different degradation and level of severity. In case of Hard, both images are degraded with mixed distortions and severity. In each setting, we randomly sample 300 image pairs. This spectrum of splits, with increasing difficulty from Easy to Hard, enables thorough evaluation of the methods to benchmark region-level understanding for distortion analysis. A few illustrative examples from each setting are presented in the appendix, see fig. 15.

| Methods | Comparison | | | | Distortion | | | | Severity | | | | Scores | |
|---|---|---|---|---|---|---|---|---|---|---|---|---|---|---|
| | A | P | R | F1 | A | P | R | F1 | A | P | R | F1 | SR | PL |
| 📄 Q-SiT (2025) | – | – | – | – | 0.16 | 0.07 | 0.05 | 0.04 | – | – | – | – | – | – |
| 📄 Q-Insight (2025) | – | – | – | – | 0.21 | 0.05 | 0.06 | 0.04 | 0.24 | 0.16 | 0.21 | 0.12 | – | – |
| 📄 DepictQA (2024b) | – | – | – | – | 0.10 | 0.06 | 0.08 | 0.05 | 0.24 | 0.17 | 0.20 | 0.10 | – | – |
| 📄 Seagull (2024c) | – | – | – | – | 0.13 | 0.10 | 0.13 | 0.08 | 0.25 | 0.25 | 0.21 | 0.18 | – | – |
| 📄 Gemma-3 27B (2025) | – | – | – | – | 0.13 | 0.09 | 0.09 | 0.07 | 0.25 | 0.26 | 0.25 | 0.22 | – | – |
| 📄 DepictQA† (2024b) | 0.33 | 0.21 | 0.18 | 0.19 | 0.22 | 0.12 | 0.09 | 0.09 | 0.30 | 0.24 | 0.23 | 0.22 | 0.18 | 0.17 |
| 🔒 GPT-5 Nano (2025) | 0.21 | 0.17 | 0.16 | 0.16 | 0.19 | 0.09 | 0.07 | 0.06 | 0.25 | 0.20 | 0.18 | 0.17 | 0.02 | 0.04 |
| 🔒 GPT-5 Mini (2025) | 0.18 | 0.19 | 0.20 | 0.15 | 0.17 | 0.10 | 0.10 | 0.09 | 0.27 | 0.22 | 0.21 | 0.20 | 0.09 | 0.13 |
| 🔒 GPT-4o (2024) | 0.16 | 0.19 | 0.16 | 0.14 | 0.15 | 0.09 | 0.07 | 0.08 | 0.23 | 0.21 | 0.18 | 0.18 | 0.06 | 0.08 |
| 🔒 Gemini 2.5 Pro (2025) | 0.12 | 0.20 | 0.14 | 0.12 | 0.11 | 0.10 | 0.07 | 0.08 | 0.17 | 0.21 | 0.13 | 0.16 | 0.10 | 0.14 |
| ⬇ Random | 0.21 | 0.20 | 0.20 | 0.19 | 0.07 | 0.07 | 0.07 | 0.06 | 0.25 | 0.25 | 0.25 | 0.24 | 0.0 | 0.0 |
| ⬇ Linear Probe | 0.43 | 0.28 | 0.21 | 0.14 | 0.30 | 0.11 | 0.09 | 0.06 | 0.24 | 0.29 | 0.26 | 0.14 | 0.22 | 0.23 |
| ⬇ Attentive Probe | 0.39 | 0.23 | 0.21 | 0.18 | 0.10 | 0.08 | 0.08 | 0.07 | 0.24 | 0.25 | 0.25 | 0.23 | 0.02 | 0.02 |
| 🐼 **PANDA** | **0.40** | **0.31** | **0.25** | **0.24** | **0.27** | **0.22** | **0.18** | **0.19** | **0.33** | **0.34** | **0.33** | **0.33** | **0.36** | **0.38** |

Table 3: **PANDABENCH Hard.** Results of different MLLMs on the Hard set. 📄 indicates open source/open-weight, and 🔒 denotes closed-source MLLMs, ⬇ stands for baselines. † indicates method is trained on PANDASET. **A**: Accuracy, **P**: Precision, **R**: Recall, **SR**: SRCC & **PL**: PLCC.

## 5 EXPERIMENTS

Given that we propose the task of learning a Distortion Graph (DG), there exist no specialist methods in literature. Hence, we consider several open-source and closed-source (frontier) MLLMs to conduct thorough experiments on the proposed PANDABENCH. Several current MLLMs such as Q-Instruct (Wu et al., 2023), Co-Instruct (Wu et al., 2024c), Janus-Pro-7B (Chen et al., 2025), and LLaVA v1.5 (Liu et al., 2024a) fail to reliably[3] perform comparative analysis at region-level. Methods like Q-Instruct (Wu et al., 2023) are not suited for multi-image tasks and are biased to the order in which multiple images are fed. While Co-Instruct (Wu et al., 2024c), Janus-Pro-7B (Chen et al., 2025), and LLaVA v1.5 (Liu et al., 2024a) can accept multiple images, but, if they are instruction tuned for distortion tasks such as Co-Instruct (Wu et al., 2024c) is, they have trouble following new instructions (Chu et al., 2025); see fig. 2 for example.

General purpose open-source MLLMs, on the other hand, suffer in distortion analysis tasks, likely due to their lack of exposure to degraded images as well as differences in training data, objectives, and scale compared to frontier models. We present a few case studies, including failure cases and discussion on their behavior, for each of these methods in appendix C.

**Open-Source MLLMs.** A few open-source/open-weights MLLMs such as Q-SiT (Zhang et al., 2025), Q-Insight (Li et al., 2025), DepictQA (You et al., 2024a), Gemma-3 27B (Team et al., 2025), and Seagull (Chen et al., 2024c) understand distortions, where Gemma-3 is general-purpose, and others are distortion-specific MLLMs. While they still can not do region-wise comparative tasks, they can classify distortion type and level of severity. We prompt open-source methods with a single region at a time overlaid with a visual marker in the form of a bounding box covering region of interest (RoI), while the rest of the image is dimmed[4], see appendix fig. 9 for sample prompt. We find that only in this manner, the output responses can be parsed and scored.

**Closed-Source MLLMs.** We also evaluate four frontier LLMs GPT-5 Nano (OpenAI, 2025), GPT-5 Mini (`gpt-5-mini-2025-08-07`) (OpenAI, 2025), GPT-4o (`gpt-4o-2024-11-20`) (Hurst et al., 2024), and Gemini 2.5 Pro (Comanici et al., 2025). Unsurprisingly, these frontier LLMs understand regions better than open-source MLLMs, and are able to perform region-wise comparative assessment. For closed-source MLLMs, we prompt them with one image pair at a time, providing each region's description and bounding box, and ask for region-wise outputs including comparison, distortion type, severity level, and a quality score within the specified range, see appendix fig. 10 for sample prompt.

**Baselines.** Lastly, we evaluate three baselines: Random, Linear Probe, and Attentive Probe. In Linear Probe, linear heads on top of DINOv2 (Oquab et al., 2023) backbone predict relations and attributes, while in Attentive Probe, a Transformer block with a cross-attention layer using a learnable query acts as the output head.

---

[3]We use 'reliably' to mean the output responses from these methods can not be scored.

[4]Passing in just the cropped region renders these methods blind due to variable size of regions.

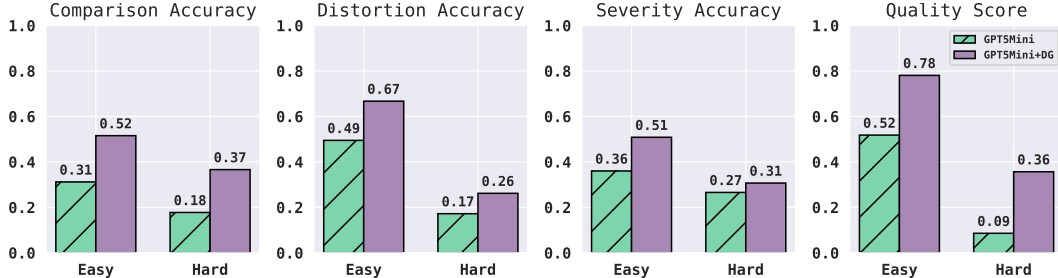

Figure 4: **Emergent Results.** Feeding predicted DG in prompt as chain-of-thought (CoT) results in improvement of $\approx 15\%$ (accuracy) in region-wise distortion understanding of GPT-5 Mini.

## 5.1 RESULTS & DISCUSSION

In tables 2, 3 and 4, we present results on the Easy, Hard, and Medium settings of PANDABENCH. For comparison, distortion, and severity type, we measure accuracy, precision, recall, and F1 score. While for quality score, we report SRCC/PLCC following Mean Opinion Score (MOS) literature (Chen et al., 2024a). In all the cases, higher is better. Across all three settings and all four tasks, our proposed method PANDA achieves the best performance. DepictQA (You et al., 2024a), despite being a much larger model (7B parameters, Vicuna v1.5 (Chiang et al., 2023) backbone), lags significantly behind PANDA, which we attribute to the absence of region-first design consideration. DepictQA, like any other open-source MLLM, suffers from context limitations, and frequently hallucinates or omits regions.

As a result, these models often fail to provide complete and faithful region-wise assessments.[5] This limitation reinforces that a structure is necessary to compactly represent pairwise information. We show that fine-tuning DepictQA[†] on PANDASET encourages region-first distortion understanding, resulting in second-best performance. A consistent trend across methods is performance decline from Easy to Hard settings, highlighting the difficulty of fine-grained distortion understanding under complex degradations. Notably, PANDA exhibits the smallest performance drop in Hard setting suggesting its efficacy. We conduct ablation studies on design choices, and discuss it in appendix B.

**Distortion & Severity Performance.** In terms of distortion and severity classification, the evaluation reveals two consistent trends. First, closed-source MLLMs achieve notably higher accuracy on distortion classification than open-source counterparts in the Easy setting, indicating a substantial performance gap. This gap, however, diminishes as the task difficulty increases, with all models converging to about a difference of less than $12\%$ under the Hard setting. Second, in severity classification, several models, including strong closed-source ones, degrade to the point of performing worse than random baseline in the Hard setting.

**Comparative & Quality Score Performance.** All of the open-source methods we compare, in tables 2, 3 and 4, struggle on region-wise comparison and quality score prediction task, including Gemma 3 (Team et al., 2025) which is a 27B parameter model. While closed-source frontier MLLMs perform better, their performance trends mirror those of distortion and severity classification. Across Easy, Medium, and Hard, every method suffers a consistent drop in accuracy, with several MLLMs degrading to near-chance performance under the Hard setting. These results underscore both the current limitations of state-of-the-art models and the value of PANDABENCH in highlighting failure modes that remain hidden under other simpler evaluation benchmarks.

## 5.2 SHOWCASE APPLICATION

In principle, a distortion graph can be learned for any task where comparative assessment informs downstream use cases. On the application front, we consider the downstream task of distortion understanding. We follow the experimental setup of Mitra et al. (2024), and adopt predicted DG in a chain-of-thought to prompt GPT-5 Mini (`gpt-5-mini-2025-08-07`) (OpenAI, 2025). Our findings on easy and hard splits of PANDABENCH, in fig. 4, indicate that coupling DG in chain-of-thought elicits emergent capabilities of LLMs in region-wise distortion understanding.

---

[5]Because of this limitation, we prompt open-source MLLMs per-region.

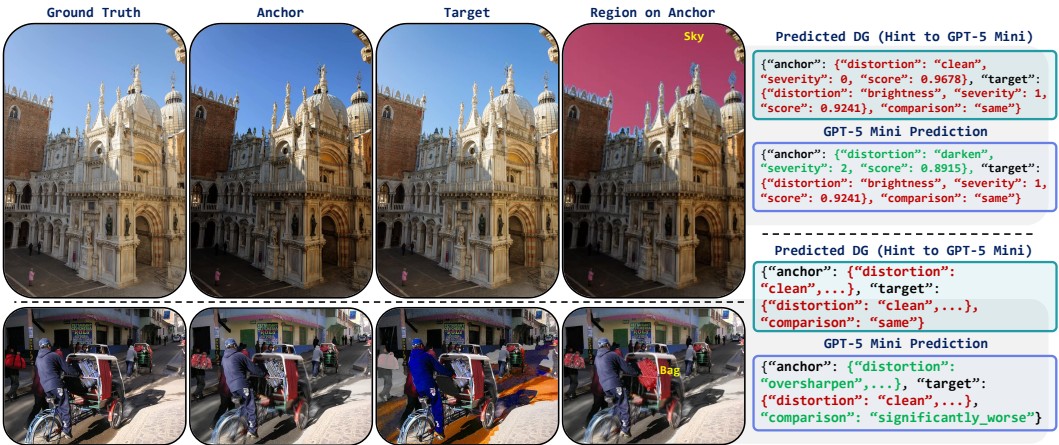

Figure 5: **Distortion Graph as Context.** Illustrative figure analyzing showcase application wherein predicted DG is fed as context to GPT-5 Mini. Top: Sample taken from PANDABENCH Easy, Bottom: Sample taken from PANDABENCH Hard. GPT-5 Mini indeed overrides the predicted DG when the pixels disagree with DG.

**Analysis of Distortion Graph as Context.** We analyze the improvement in fig. 4 and evaluate whether GPT-5 Mini simply copies the DG predictions verbatim. We explicitly instruct GPT-5 Mini to treat DG as a hint and to fall back to the input pixels whenever a conflict arises, and we find that it often does override DG when the visual evidence disagrees, see fig. 5. For example, in the PANDABENCH Easy sample (top row of fig. 5), DG incorrectly predicts the distortion type on the anchor region as *clean*, while GPT-5 Mini correctly identifies it as *darken*; this requires actually comparing the anchor and target images rather than copying the graph. Likewise, in the PANDABENCH Hard sample, DG mislabels the bag region as *clean*, but GPT-5 Mini correctly infers *oversharpen*. We observe a few such cases where GPT-5 Mini corrects DG using pixel evidence.

On the flip side, when there is little or no contradictory signal in the pixels, GPT-5 Mini tends to trust DG. In the Easy example, for instance, the target sky region is predicted as *brightness* by DG, and GPT-5 Mini repeats this label even though the ground-truth degradation is *clean*. We emphasize that this behavior is precisely the intended usage: DG acts as an additional structured cue that the MLLM may either leverage directly or override when its own visual understanding disagrees.

## 6 CONCLUSION

In this work, we introduced the task of Distortion Graph (DG), a region-grounded topological representation for pairwise image assessment. We argued that structured representations like DG provide an efficient, compact, and interpretable means for comparative evaluation. To support this task, we contributed PANDABENCH, a benchmark for assessing the region-level distortion understanding of MLLMs, and demonstrated that current open-source models exhibit clear gaps in region-aware analysis. Our experiments showed that either training on PANDASET or prompting with DG as part of reasoning chains substantially improves region-wise assessment. We hope this work motivates further exploration of region-first representations for distortion understanding and establishes DG as a useful data structure for fine-grained comparative reasoning.

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

# TECHNICAL APPENDICES

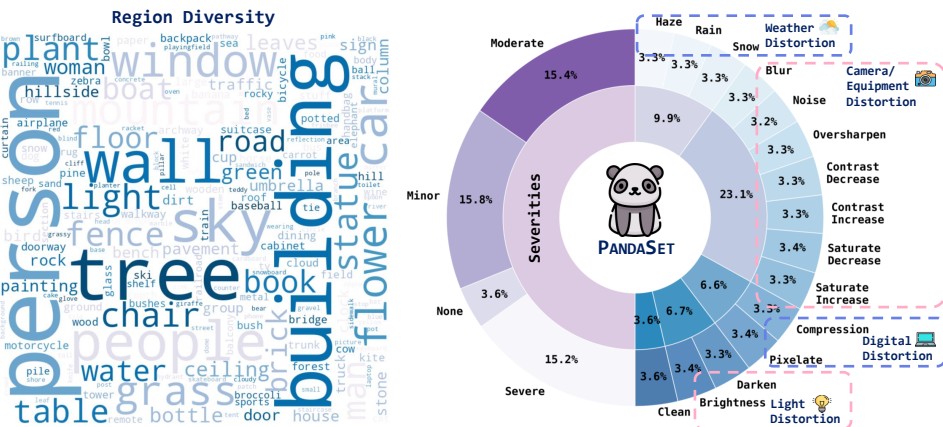

Figure 6: **PANDASET Summary.** Left: A word cloud of region names indicating diversity of the objects in images. Right: A region-wise summary of PANDASET in terms of distortions & severity. All 15 of the distortions are uniformly distributed across the regions, and we broadly categorize the distortions in super categories: 🌤 *weather,* 📷 *camera/equipment,* 💻 *digital,* 💡 *light, and clean.*

| Methods | Comparison | | | | Distortion | | | | Severity | | | | Scores | |
|---|---|---|---|---|---|---|---|---|---|---|---|---|---|---|
| | A | P | R | F1 | A | P | R | F1 | A | P | R | F1 | SR | PL |
| Q-SiT (2025) | – | – | – | – | 0.18 | 0.09 | 0.07 | 0.05 | – | – | – | – | – | – |
| Q-Insight (2025) | – | – | – | – | 0.19 | 0.08 | 0.07 | 0.05 | 0.21 | 0.19 | 0.22 | 0.12 | – | – |
| DepictQA (2024b) | – | – | – | – | 0.13 | 0.09 | 0.11 | 0.07 | 0.27 | 0.14 | 0.20 | 0.11 | – | – |
| Seagull (2024c) | – | – | – | – | 0.19 | 0.18 | 0.16 | 0.13 | 0.28 | 0.27 | 0.22 | 0.20 | – | – |
| Gemma-3 27B (2025) | – | – | – | – | 0.21 | 0.21 | 0.17 | 0.13 | 0.26 | 0.27 | 0.26 | 0.23 | – | – |
| DepictQA† (2024b) | 0.32 | 0.30 | 0.25 | 0.27 | 0.47 | 0.56 | 0.41 | 0.46 | 0.43 | 0.40 | 0.34 | 0.36 | 0.44 | 0.42 |
| GPT-5 Nano (2025) | 0.25 | 0.21 | 0.22 | 0.20 | 0.26 | 0.20 | 0.16 | 0.16 | 0.28 | 0.26 | 0.23 | 0.21 | 0.18 | 0.24 |
| GPT-5 Mini (2025) | 0.22 | 0.22 | 0.25 | 0.19 | 0.31 | 0.29 | 0.24 | 0.24 | 0.32 | 0.28 | 0.25 | 0.25 | 0.29 | 0.34 |
| GPT-4o (2024) | 0.19 | 0.20 | 0.21 | 0.17 | 0.28 | 0.25 | 0.21 | 0.22 | 0.27 | 0.26 | 0.22 | 0.22 | 0.28 | 0.33 |
| Gemini 2.5 Pro (2025) | 0.14 | 0.21 | 0.18 | 0.13 | 0.24 | 0.34 | 0.20 | 0.23 | 0.24 | 0.29 | 0.19 | 0.23 | 0.31 | 0.34 |
| Random | 0.20 | 0.20 | 0.20 | 0.19 | 0.07 | 0.07 | 0.07 | 0.06 | 0.25 | 0.25 | 0.25 | 0.25 | 0.00 | 0.00 |
| Linear Probe | 0.38 | 0.29 | 0.21 | 0.14 | 0.24 | 0.16 | 0.09 | 0.06 | 0.28 | 0.27 | 0.26 | 0.15 | 0.17 | 0.20 |
| Attentive Probe | 0.39 | 0.34 | 0.29 | 0.29 | 0.24 | 0.25 | 0.24 | 0.23 | 0.26 | 0.25 | 0.26 | 0.24 | 0.16 | 0.21 |
| PANDA | **0.44** | **0.44** | **0.38** | **0.40** | **0.52** | **0.55** | **0.50** | **0.52** | **0.47** | **0.49** | **0.47** | **0.48** | **0.56** | **0.61** |

Table 4: **PANDABENCH Medium.** Results of different MLLMs on the Medium set. 💬🔓 indicates open source/open-weight, and 🔒 denotes closed-source MLLMs, ⬇ stands for baselines. † indicates method is trained on PANDASET. **A**: Accuracy, **P**: Precision, **R**: Recall, **SR**: SRCC & **PL**: PLCC.

## A MOTIVATION

It is well-studied and understood that a proper structured representation enables many of the visual intelligence tasks in humans (Swoyer, 1991) and machines (Chiou, 2022) alike: spatial understanding (Zhang et al., 2024; Lorenz et al., 2025), visual planning (Gu et al., 2024), video reasoning (Wang et al., 2025), and even similarity comparisons in visual input in the case of humans (Hodgetts et al., 2023). One of the fundamentals of human decision making are pairwise comparisons. Classic psychophysics has established that pairwise comparison produces reliable and interpretable perceptual scales compared to absolute ratings (Saffir, 1937), and enable principled selection. It is, thus, natural to consider if a structured representation can also aid such comparative decisions. We argue that our proposed Distortion Graph (DG) offers a general-purpose structure and acts as a scaffold towards these decisions.

**DG as a General Comparative Formalism.** Unlike conventional approaches that rely on holistic embeddings or scalar quality scores, DG decomposes perceptual differences into object-anchored nodes, attribute descriptors, and explicit comparative relations across paired inputs. This formalism provides several advantages. First, DG offers a general abstract: the same formalism that encodes region-wise distortions in images can naturally extend to other setups and modalities, such

as pose differences in paired videos for video action differencing task (Burgess et al., 2025), region grounded differences for forgery detection (Sun et al., 2025; Xu et al., 2023), pairwise CT scan assessment (Hoeijmakers et al., 2024), benchmarking image signal processors (ISPs) (Yunfan et al., 2024), compressing redundant frames in memory based on similarity (Reza et al., 2025), etc. Second, DG is inherently interpretable, i.e., each edge explicitly localizes where and how two inputs diverge, enabling fine-grained analysis that opaque embedding distances can not provide. Third, by relying on a fixed predicate set (such as for comparisons in image quality assessment), DG encourages compositional reasoning. Finally, the structured format of DG makes it a natural scaffold for multimodal large language models (MLLMs), which often benefit from symbolic context to reduce hallucination and improve grounding. DG also lifts the requirement of Supervised Finetuning (SFT) the MLLMs on a fixed distortion set, and general purpose LLMs, with better instruction following abilities, can be coupled as is. We illustrate this in fig. 4.

Taken together, these properties suggest that DG offers structured reasoning over differences in the visual input, and is a step toward a unifying comparative formalism for assessment tasks.

## B  ABLATION STUDIES

Linear and Attentive probes in tables 2, 3 and 4 serve as ablations of proposed decoder in PANDA architecture. Their performance drop shows that DINOv2 (Oquab et al., 2023) features alone are insufficient, and that the decoder is crucial for enabling each region to retrieve complementary information from its pair to learn distortion relationships and attributes. We conduct two more ablations on feature extractors and the number of Transformer blocks in the decoder. Our default backbone, DINOv2 (ViT-s), yields features of 384 dimensions, and we ablate DINOv2 (ViT-b) and SigLip (Zhai et al., 2023) with a dimension size of 768. By default, we adopt four Transformer blocks in the decoder of PANDA. We also ablate its sufficiency by considering two and six blocks as variations. Figure 7 shows that design choices in PANDA are an optimal balance in network size and performance.

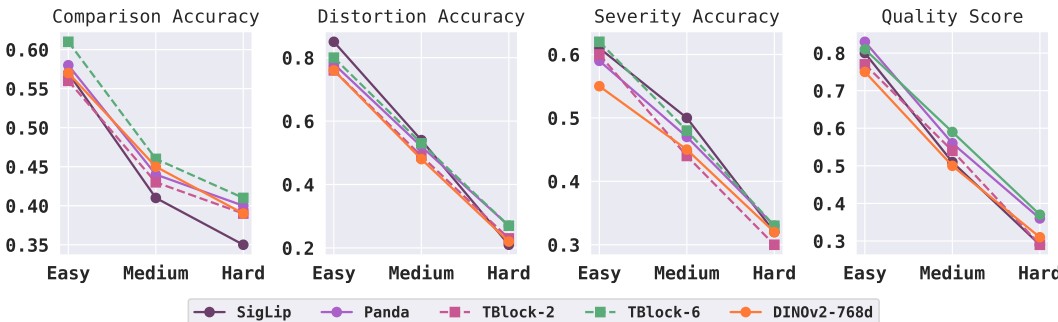

Figure 7: **Design Choice Ablation.** Accuracy comparison of different design choices: backbone feature extractors (solid line) and Transformer blocks (dotted line). PANDA maintains balance in size, performance, and efficiency.

**Whole Image vs. Region-Wise.**   Our findings indicate that MLLMs performance is dependent on the granularity of decision-making. If whole image, i.e., global view, is considered, the performance is non-trivial, but the same MLLMs suffer when reasoning over regions due to (i) lack of region-wise design considerations and (ii) rigidity induced by SFT on a fixed set of distortions/settings. PAND-ABENCH illustrates this struggle in tables 2, 3 and 4. Recall that in the easy split of PANDABENCH, a single distortion afflicts the whole image. While we reason over regions even in that split, it is possible to query an MLLM at the whole-image level for the distortion classification task. In table 5, we report results of an ablation study towards this end with DepictQA (You et al., 2024a). DepictQA achieves low but non-trivial performance when asked to classify distortions at the whole-image level. In contrast, when queried region-wise, the performance sharply drops to chance.

**Cost to Query MLLMs.**   We analyze the cost to query MLLMs, both in terms of computation and monetary value. We also report the configuration of the MLLM, the LLM and vision tower,

| Type | Distortion Classification | | | |
|------|:---:|:---:|:---:|:---:|
| | Accuracy ↑ | Precision ↑ | Recall ↑ | F1 ↑ |
| **Region Wise** | 0.15 | 0.11 | 0.13 | 0.10 |
| **Whole Image** | 0.26 | 0.32 | 0.28 | 0.26 |

Table 5: **Region-wise vs. Whole Image.** Comparison of DepictQA (You et al., 2024a) across two different evaluation setups: whole image and region-wise for distortion classification task.

| Method | LLM | Vision Tower | Compute Cost / Image Pair (secs) | Monetary Value / Image Pair (USD) | Parameters (Billion) |
|--------|-----|-------------|:---:|:---:|:---:|
| Q-SiT (2025) | Qwen2 LLM | SigLIP | 57.74 | N/A | 7 |
| Q-Insight (2025) | Qwen2.5 LLM | Qwen2.5 | 274.74 | N/A | 7 |
| DepictQA (2024a) | Vicuna v1.5 | CLIP | 245.42 | N/A | 7 |
| Seagull (2024c) | Vicuna v1.1 | CLIP | 38.19 | N/A | 7 |
| GPT-5 Nano (2025) | Proprietary | Proprietary | N/A | 0.002 | Proprietary |
| GPT-5 Mini (2025) | Proprietary | Proprietary | N/A | 0.007 | Proprietary |
| GPT-4o (2024) | Proprietary | Proprietary | N/A | 0.028 | Proprietary |
| PANDA | N/A | DINOv2 | 3.53 | N/A | 0.028 |

Table 6: **Cost Analysis.** A summary of compute and monetary costs of different methods computed for a single image pair with 14 regions.

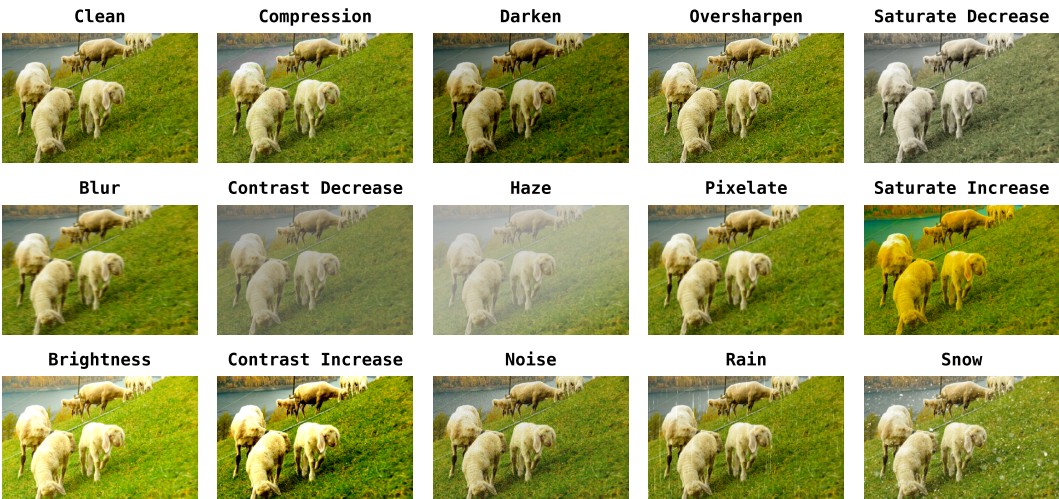

Figure 8: **All Distortion Types.** We visualize all 15 different distortion types on the same image taken from PANDASET. Each distortion degrades the image differently. Some distortions ruin the perceptual quality of the image more than others (e.g., haze, contrast decrease).

along with model size in parameters in table 6. For closed-source models, we do not compare the compute cost since they are exposed through an API. While for open-source models, we do not report monetary value given it is hard to estimate. All costs are reported on a single NVIDIA v100 GPU (32GB) with batch size 1 and an image pair with 14 regions. Notably, PANDA is significantly cheaper in terms of computational cost and parameters.

## B.1 DISTORTION GRAPH FORMALISM AGGREGATES TO WHOLE-IMAGE

We argue that region-first distortion analysis aggregates to and complements whole-image assessment naturally. We consider the instant rating task wherein given two images, the task is to rank which better image is perceptually superior. We adopt the KADID10k dataset (Lin et al., 2019). Due to lack of region information in KADID10k, we query Segment Anything (SAM) (Kirillov et al., 2023) to generate region masks. We do zero-shot inference with PANDA trained on PANDASET, and report performance on the ranking task using the predicted quality score or the comparative rela-

| Method | Ranking Accuracy ↑ | Inference Time ↓ |
|---|---|---|
| Q-Insight (Li et al., 2025) | 0.6970 | 8 hours |
| GPT-5 Mini (OpenAI, 2025) | 0.8472 | N/A |
| 🐼 PANDA (ZS) Score Based | 0.7883 | 4 mins |
| 🐼 PANDA (ZS) Predicate Based | 0.7690 | 4 mins |

Table 7: **Whole-Image Instant Ranking.** DG (generated by PANDA) aggregates naturally to whole-image assessment.

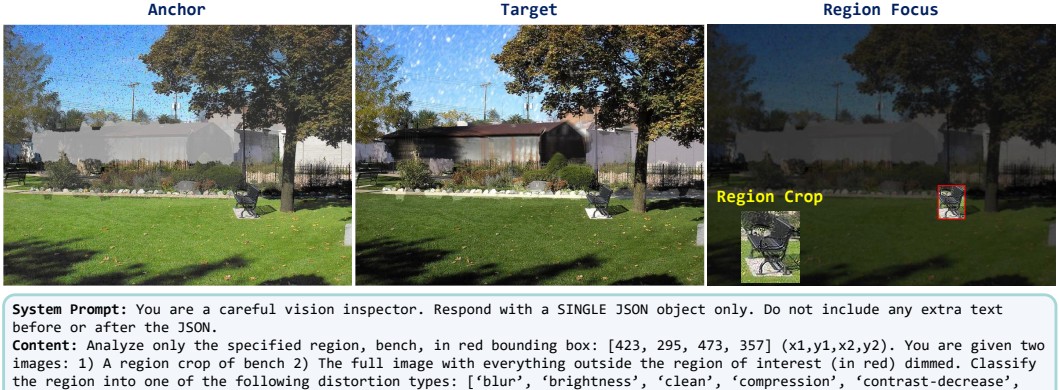

**System Prompt:** You are a careful vision inspector. Respond with a SINGLE JSON object only. Do not include any extra text before or after the JSON.
**Content:** Analyze only the specified region, bench, in red bounding box: [423, 295, 473, 357] (x1,y1,x2,y2). You are given two images: 1) A region crop of bench 2) The full image with everything outside the region of interest (in red) dimmed. Classify the region into one of the following distortion types: ['blur', 'brightness', 'clean', 'compression', 'contrast-decrease', 'contrast-increase', 'darken', 'haze', 'noise', 'oversharpen', 'pixelate', 'rain', 'saturate-decrease', 'saturate-increase', 'snow'], and one of the following severity types: ['minor', 'moderate', 'none', 'severe']. For example: {"distortion": ..., "severity": ...} and replace ... with appropriate distortion and severity type.

Figure 9: **Prompt Type (a).** A template of prompt for open-source MLLMs. The tags for keywords like image, input, user, assistant, output, etc. that each method requires are added as necessary.

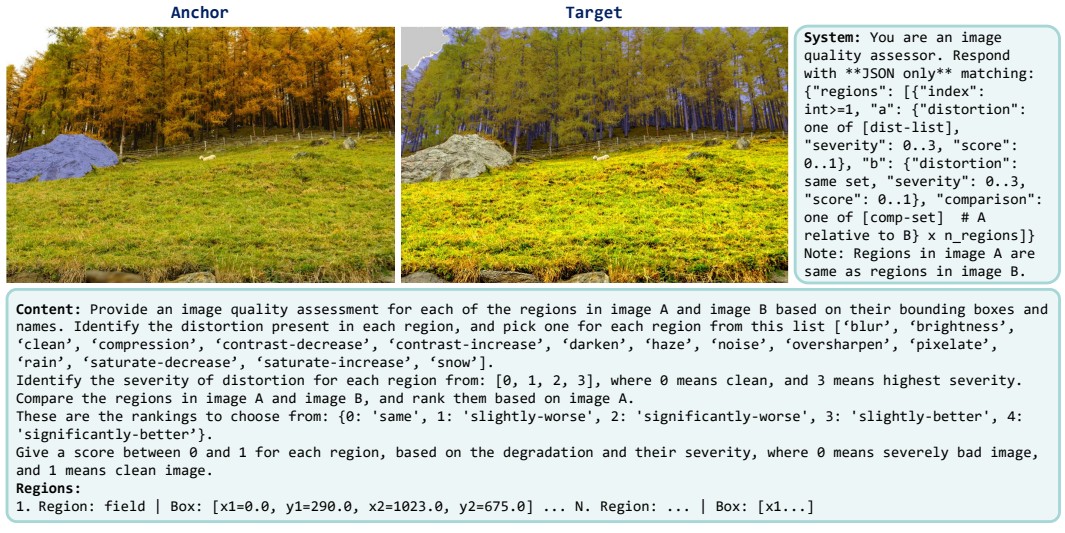

**System:** You are an image quality assessor. Respond with **JSON only** matching: {"regions": [{"index": int>=1, "a": {"distortion": one of [dist-list], "severity": 0..3, "score": 0..1}, "b": {"distortion": same set, "severity": 0..3, "score": 0..1}, "comparison": one of [comp-set]  # A relative to B} x n_regions]} Note: Regions in image A are same as regions in image B.

**Content:** Provide an image quality assessment for each of the regions in image A and image B based on their bounding boxes and names. Identify the distortion present in each region, and pick one for each region from this list ['blur', 'brightness', 'clean', 'compression', 'contrast-decrease', 'contrast-increase', 'darken', 'haze', 'noise', 'oversharpen', 'pixelate', 'rain', 'saturate-decrease', 'saturate-increase', 'snow'].
Identify the severity of distortion for each region from: [0, 1, 2, 3], where 0 means clean, and 3 means highest severity.
Compare the regions in image A and image B, and rank them based on image A.
These are the rankings to choose from: {0: 'same', 1: 'slightly-worse', 2: 'significantly-worse', 3: 'slightly-better', 4: 'significantly-better'}.
Give a score between 0 and 1 for each region, based on the degradation and their severity, where 0 means severely bad image, and 1 means clean image.
**Regions:**
1. Region: field | Box: [x1=0.0, y1=290.0, x2=1023.0, y2=675.0] ... N. Region: ... | Box: [x1...]

Figure 10: **Prompt Type (b).** A template of prompt for closed-source MLLMs. Frontier methods have superior instruction following ability, and can reason about the regions from the prompt.

tionships (predicates). PANDA was not originally trained to provide whole-image ranking, and we directly use predicted relationship predicates or region-wise scores with a naive control logic, e.g., if more regions in image A are better (or score higher), then image A is better, to compute the ranking accuracy in table 7. Distortion graph (DG) naturally extends to whole-image assessment.

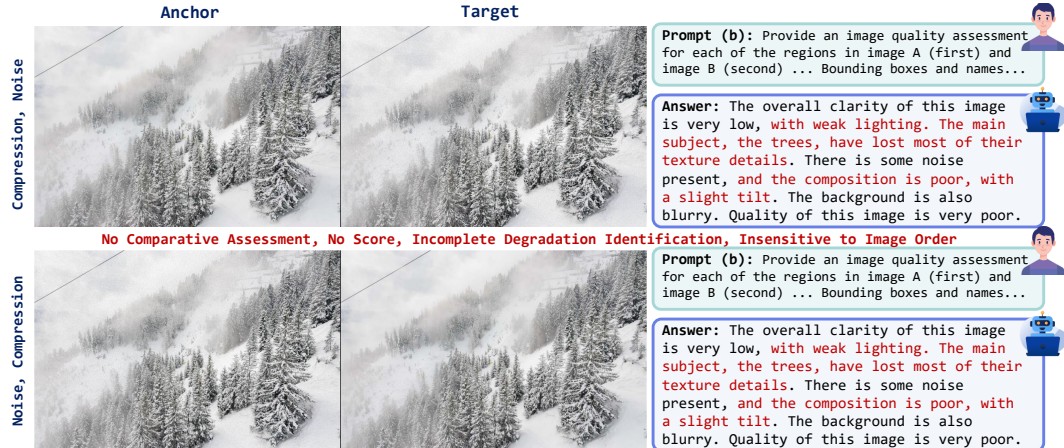

Figure 11: **Q-Instruct Behavior.** An illustration of output from Q-Instruct (Wu et al., 2023) when prompted for multiple instructions. It is insensitive to the order of image, even when explicitly specified, misses degradation, struggles to follow instruction, and repeats irrelevant information.

## B.2 GENERALITY OF DISTORTION GRAPH REPRESENTATION

Although PANDABENCH is specifically designed for region-first, pairwise comparison, we find that the Distortion Graph formalism generalizes beyond our protocol and naturally aggregates to whole-image ranking on standard human-annotated MOS benchmarks. In table 7, we evaluate PANDA on KADID10k (Lin et al., 2019) by using the predicted DG to rank image pairs and comparing against ground-truth Mean Opinion Scores. Using the DG's scalar score attribute as the ranking signal yields an accuracy of 78.83%, while using the comparative predicate over regions yields 76.90%, indicating that a model trained on TOPIQ-derived region labels can still produce

| Methods | Accuracy ↑ |
|---|---|
| mPLUG-Owl2 (2024) | 48.5 |
| LLaVA-1.6 (2024b) | 57.0 |
| Q-Instruct (2024b) | 55.0 |
| 🐼 PANDA (Predicate) | **78.4** |
| 🐼 PANDA (Score) | **77.8** |

Table 8: **Generality of DG Representation on TID2013.**

image-level rankings that align well with human judgments. To further test robustness, we repeat the same protocol on TID2013 (Ponomarenko et al., 2015), another widely used IQA dataset with diverse distortion types and human-annotated MOS, see table 8. Here, PANDA again achieves strong performance, 77.8% accuracy when ranking with DG scores and 78.4% with DG predicates, outperforming MLLM-based baselines such as mPLUG-Owl2 (Ye et al., 2024), LLaVA-1.6 (Liu et al., 2024b), and Q-Instruct (Wu et al., 2024b). These results support that DG is not merely a dataset-specific construct for PANDABENCH but a generally useful representation that (i) can be collapsed into reliable whole-image rankings and (ii) transfers effectively to real-distortion, human-labeled benchmarks.

## C ANALYSIS OF MULTIMODAL LARGE LANGUAGE MODELS

In this section, we present analysis of the behavior of different Multimodal Large Language Models (MLLMs), both open-source and closed-source, considered in this manuscript on region-wise distortion understanding. We divide the analysis into three settings based on the results in tables 2, 3 and 4 (i) *methods considered, but not reported*, (ii) *open-source methods considered, and reported*, and (iii) *closed-source methods considered, and reported*. As discussed in section 5, we divide the prompt templates into two categories: (a) *open-source prompts* and (b) *closed-source prompts*. This is because open-source MLLMs fail to provide an answer since they can either consider the whole image or just one region at a time. In contrast, closed-source models have superior instruction following abilities and can predict distortion and severity type, along with comparative assessment and quality score for all the regions in the image pair. We present samples of both prompt types (a)

in fig. 9 and (b) in fig. 10. Note that in prompt (a) fig. 9, three images are fed to the MLLM, namely the anchor or the target, region focus and crop of the region for the respective image.

## C.1 METHODS CONSIDERED, NOT REPORTED

Recall that among open-source methods, we also explored Q-Instruct (Wu et al., 2023) (LLaVA v1.5 (Liu et al., 2024a)), Co-Instruct (Wu et al., 2024c), and Janus-Pro-7B (Chen et al., 2025), but did not report in tables 2, 3 and 4 due to unreliability in their respective outputs (see section 5). We detail their respective behavior on PANDABENCH, illustrate with sample outputs, and briefly discuss the reasons.

**Q-Instruct.** Q-Instruct (Wu et al., 2023), a distortion MLLM, is designed for single image distortion analysis, but not for comparative assessment. A common workaround for multi-image inputs is to stack two images before prompting, as in Fu et al. (2024). However, we observe that Q-Instruct does not consistently adapt to the change in order of stacked images. For example, when the left image (anchor) is labeled as noise and the right image (target) as compression, flipping their order does not necessarily flip the predicted degradations. This order-dependence makes the outputs unreliable for comparative tasks, so we do not report Q-Instruct results. An illustrative example of such behavior is presented in fig. 11.

**Co-Instruct.** Co-Instruct (Wu et al., 2024c) is a distortion MLLM designed for multi-image comparisons. Given two (or more) input images, it can describe the quality of the images and compare or rank them. Our experiments show that it struggles with the multi-instruction setting required by PANDABENCH. In practice, it often omits regions due to output length limits or repeats the same region until tokens are exhausted. Moreover, it fails to generalize to tasks outside its supervised fine-tuning regime such as the four region-level tasks in PANDABENCH (distortion classification, severity classification, predicate identification, and quality score prediction). An example of this behavior is shown in fig. 12.

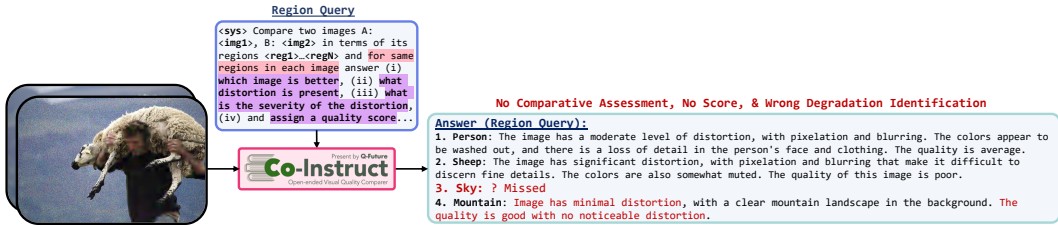

Figure 12: **Co-Instruct Behavior.** An illustration of output from Co-Instruct (Wu et al., 2024c) when prompted for multiple instructions. It fails to perform comparative assessment, frequently misses regions, and struggles with instruction following.

**Janus-Pro-7B.** Unlike Q-Instruct (Wu et al., 2023) and Co-Instruct (Wu et al., 2024c), Janus-Pro-7B (Chen et al., 2025) is a general-purpose open-source MLLM designed for various multimodal tasks. However, when applied to low-level vision, its effectiveness remains limited, reflecting a broader trend among open-source general-purpose MLLMs. We prompt Janus-Pro-7B with prompt type (a), see fig. 9, on PANDABENCH Medium split, and observe that it almost invariably predicts the same distortion and severity label for nearly every region. This indicates that the method lacks reliable distortion understanding ability and is insensitive to degradation inherent in the inputs, making its performance uninformative. The output is similar to the one presented in fig. 13, except for each region predicted distortion and severity class are *clean*.

## C.2 OPEN-SOURCE METHODS CONSIDERED, REPORTED

As discussed earlier, in open-source methods, we consider Q-Insight (Li et al., 2025), Q-SiT (Zhang et al., 2025), Gemma 3-27B (Team et al., 2025), Seagull (Chen et al., 2024c), and DepictQA (You et al., 2024a). Other than Gemma 3-27B, all other methods are distortion-specific MLLMs. Given that images in PANDABENCH have variable number of regions, and these methods are limited in the number of new tokens they can generate, we adopt prompt type (a), see fig. 9. We modify the prompt with special tokens as necessary for each method. Our findings indicate that their perfor-

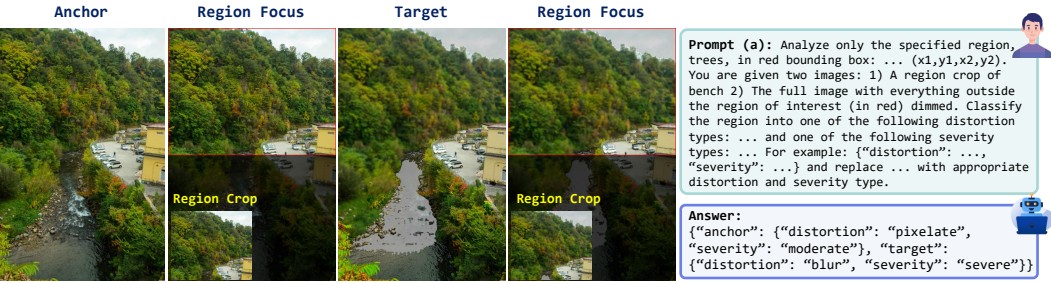

Figure 13: **Open-Source MLLM Prompt/Output.** A representative example of prompt type (a) along with output for all open-source MLLMs evaluated in this work.

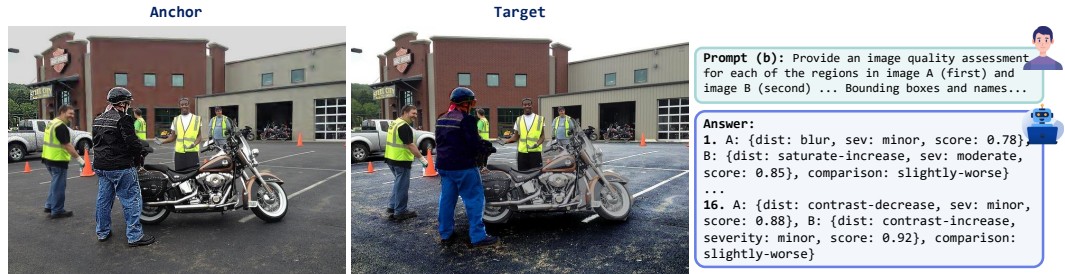

Figure 14: **Closed-Source MLLM Prompt/Output.** A representative example of prompt type (b) along with output for all closed-source MLLMs evaluated in this work. Note that, in this example, the image has 16 regions.

mance is generally limited, especially on Hard split of PANDABENCH indicating a broader trend in lack of region-wise image understanding towards distortion analysis. While Seagull (Chen et al., 2024c) is a region-first method, it struggles with following instructions and can not do comparative assessment. Since the maximum new generated tokens are limited, we query all of these methods region-wise for each image pair, i.e., a separate forward pass for each region in each pair. Hence, we do not report performance on comparative task. For quality score prediction, we observe that the outputs are largely static, varying only in coarse steps of 0.25. This behavior leads to low pearson linear correlation coefficient (PLCC) and spearman rank-order correlation coefficient (SRCC), and thus we consider the results unreliable and do not report them. A typical prompt/output example representative of open-source methods is presented in fig. 13.

### C.3 CLOSED-SOURCE METHODS CONSIDERED, REPORTED

We evaluate four frontier closed-source LLMs on PANDABENCH, namely GPT-5 Nano (OpenAI, 2025), GPT-5 Mini (gpt-5-mini-2025-08-07) (OpenAI, 2025), GPT-4o (gpt-4o-2024-11-20) (Hurst et al., 2024), and Gemini 2.5 Pro (Comanici et al., 2025). Compared to open-source models, closed-source frontier MLLMs exhibit stronger instruction-following abilities and are not constrained by output length. In practice, we find that a limit of 8192 tokens is sufficient to cover all regions in PANDABENCH images. Based on this observation, we employ prompt template type (b) (see fig. 10) for these models. For each split, we conduct three independent runs per method and report the mean performance across runs in tables 2, 3 and 4. We additionally report corresponding standard deviations in table 9 for accuracy metric on all four tasks of PANDABENCH. A typical prompt/output example representative of closed-source methods is presented in fig. 14.

### D ILLUSTRATION OF PANDABENCH

PANDABENCH comprises three splits, Easy, Medium, and Hard, taken from the test set of proposed dataset PANDASET. From Easy to Hard, the task difficulty progressively increases. Recall that in Easy split, both images in each pair are degraded by the same distortion type applied uniformly

| Standard Deviation | | GPT-5 Nano | GPT-5 Mini | GPT-4o | Gemini 2.5 Pro |
|---|---|---|---|---|---|
| **PANDABENCH Hard** | Distortion | 0.0077 | 0.0016 | 0.0075 | 0.0029 |
| | Severity | 0.0086 | 0.0025 | 0.0083 | 0.0033 |
| | Comparison | 0.0065 | 0.0027 | 0.0105 | 0.0060 |
| | SRCC | 0.0099 | 0.0076 | 0.0067 | 0.0070 |
| | PLCC | 0.0097 | 0.0050 | 0.0077 | 0.0081 |
| **PANDABENCH Medium** | Distortion | 0.0025 | 0.0076 | 0.0115 | 0.0104 |
| | Severity | 0.0084 | 0.0015 | 0.0040 | 0.0106 |
| | Comparison | 0.0059 | 0.0055 | 0.0138 | 0.0062 |
| | SRCC | 0.0116 | 0.0054 | 0.0078 | 0.0079 |
| | PLCC | 0.0103 | 0.0102 | 0.008 | 0.0077 |
| **PANDABENCH Easy** | Distortion | 0.0061 | 0.0239 | 0.0271 | 0.0090 |
| | Severity | 0.0065 | 0.0122 | 0.0183 | 0.0094 |
| | Comparison | 0.0088 | 0.0121 | 0.0118 | 0.0083 |
| | SRCC | 0.0207 | 0.0100 | 0.0052 | 0.0063 |
| | PLCC | 0.0309 | 0.0072 | 0.0064 | 0.0055 |

Table 9: **Standard Deviation on Accuracy Metric.** The standard deviation for accuracy metric on all four tasks of PANDABENCH computed over three independent runs of each method. We do not report standard deviation for precision, recall and F1 score for brevity, but they follow similar trends.

across all regions, but with differing levels of severity. In the Medium split, one image is consistently degraded by a single distortion across all regions, while its paired image exhibits region-wise distortions sampled randomly from the full distortion set. We illustrate the three splits in fig. 15. Notice how in PANDABENCH Hard, each region has different degradation, e.g., the ground in middle image (last row) has noise, while it is free of noise in its pair on left (last row).

## E  HYPERPARAMETER SENSITIVITY ANALYSIS

The optimization objective of PANDA is $\mathcal{L} = \lambda_1 L_{CE}^{\text{rel}} + \lambda_2 L_{CE}^{\text{dist}} + \lambda_3 L_{CE}^{\text{sev}} + \lambda_4 L_1^{\text{score}}$. We search for the value of each $\lambda \in \{0.01, 0.1, 1.0, 5.0\}$ using cross-validation. A set of baseline values (obtained with each $\lambda$ set as 1.0) serve as an indication for early stopping, and we only train for full 30 epochs if a particular combination is better than the baseline. We set the final objective as $\mathcal{L} = 0.1 \times L_{CE}^{\text{rel}} + 1.0 \times L_{CE}^{\text{dist}} + 0.1 \times L_{CE}^{\text{sev}} + 1.0 \times L_1^{\text{score}}$, where $\lambda_1 = 0.1, \lambda_2 = 1.0, \lambda_3 = 0.1, \lambda_4 = 1.0$. Note that each $\lambda$ is common for both its respective heads, i.e., for both anchor and target the same $\lambda$ is used. We present results of different runs in fig. 16 wherein each grey point denotes an experiment that performed significantly worse and we label top five settings with colored $\times$ mark. PANDA is trained for 30 epochs with a batch size of 6 on $8\times$ NVIDIA v100 32GB GPUs, and it processes all regions for an image pair simultaneously. PANDA takes around 1.5 days to train for all 30 epochs, and in inference we employ one NVIDIA v100 32GB GPU. For learning rate, we swept through $\{1e^{-2}, 1e^{-4}, 1e^{-6}\}$, and found that $1e^{-4}$ best balanced speed of optimization and convergence of optimization procedure. As shown in fig. 16, performance remains largely consistent across most hyperparameter configurations, as shown by the tight cluster of grey points around similar performance values, with only a few extreme combinations yielding noticeable differences. This suggests that PANDA is not overly sensitive to hyperparameter selection, and that reasonable choices are sufficient for stable performance.

## F  DISTORTION GRAPHS

We discuss a few limitations of this work, PANDA, detail directions for future work, and present a reproducibility and data statement. We also present a sample of a dense distortion graph generated from an image pair with several regions in fig. 17.

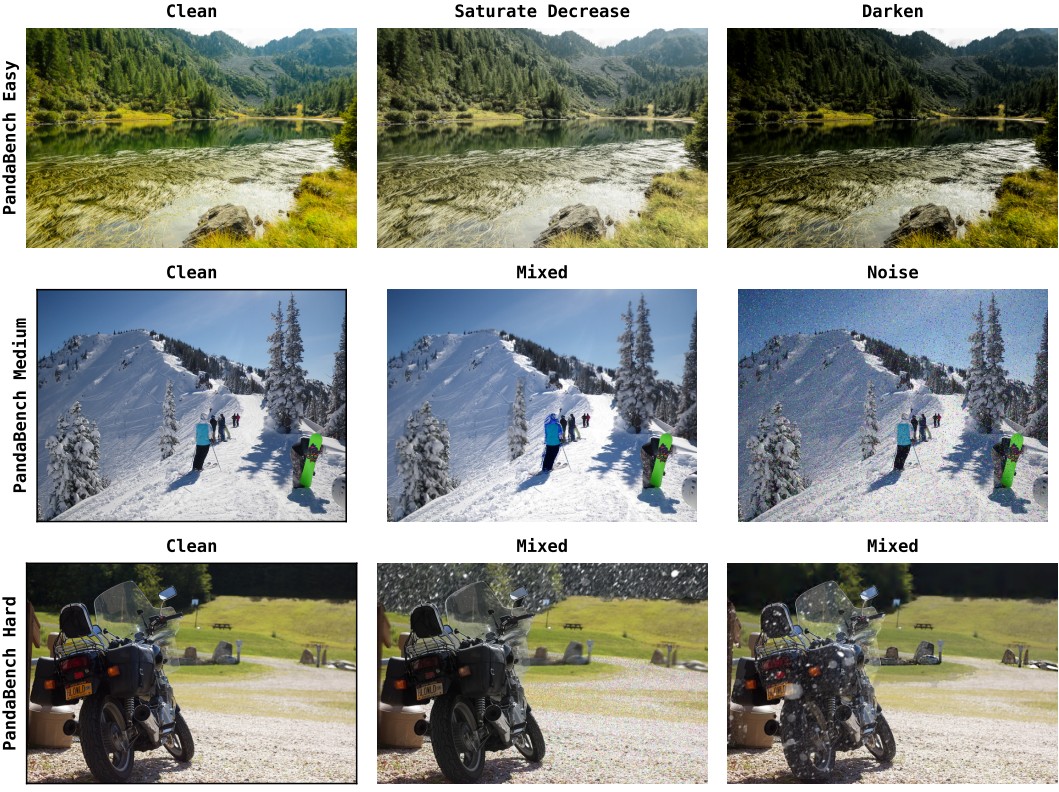

Figure 15: **PANDABENCH.** Representative samples from Easy, Medium, and Hard splits of PAND-ABENCH. In Easy split, only one distortion afflicts the entire image (and its regions, but with varied severity), while in Medium, mixed has region-wise distortions (see the person in blue jacket). In Hard split, the distortion varies by region in both images (see ground, bike, trees, etc.). Taken together, they represent a spectrum of difficulty with subtle degradations inherent in the image to complicated region-wise distortions.

## F.1 LIMITATIONS & FUTURE WORK

PANDA serves as a minimal yet necessary baseline for learning the proposed distortion graph task. While DG provides a complete intra- (semantics) and inter-image (distortions) representation, PANDA remains intentionally simple and leaves room for improvement, particularly in handling complex regions. An interesting future direction would be to employ DG as a separate intermediate step where the graph is generated as part of the reasoning chains for region-wise distortion reasoning before final answer is generated. Furthermore, as DG provides a general formalism for comparative reasoning, an interesting avenue for future work is to extend it beyond distortion analysis to broader comparative tasks in vision and multimodal settings.

Another potential avenue for future work pertains to the construction of PANDASET. While its scenes are natural, and we preserve the real-world ISP distortions, from Seagull-100w (Chen et al., 2024c), when they overlap with chosen distortion category, e.g., noise, blur, etc., the remaining distortions are synthetic following (You et al., 2024b;a). This design is intentional because control-lable distortions are what makes it possible to (i) assign deterministic region-level quality scores, (ii) match regions with comparative labels, and (iii) systematically vary difficulty from Easy to Hard in PANDABENCH. We show that our design choices are aligned with human preferences, see tables 7 and 8, but it is possible that comparative relations inherit underlying IQA model's, TOPIQ (Chen et al., 2024a), perceptual biases. More broadly, the lack of a large-scale, region-grounded, real-world comparative dataset in the literature is a key limiting factor. Building PANDASET with human-annotated region-level comparative relations at similar scale would require a substantial annotation effort, which we leave to the future work.

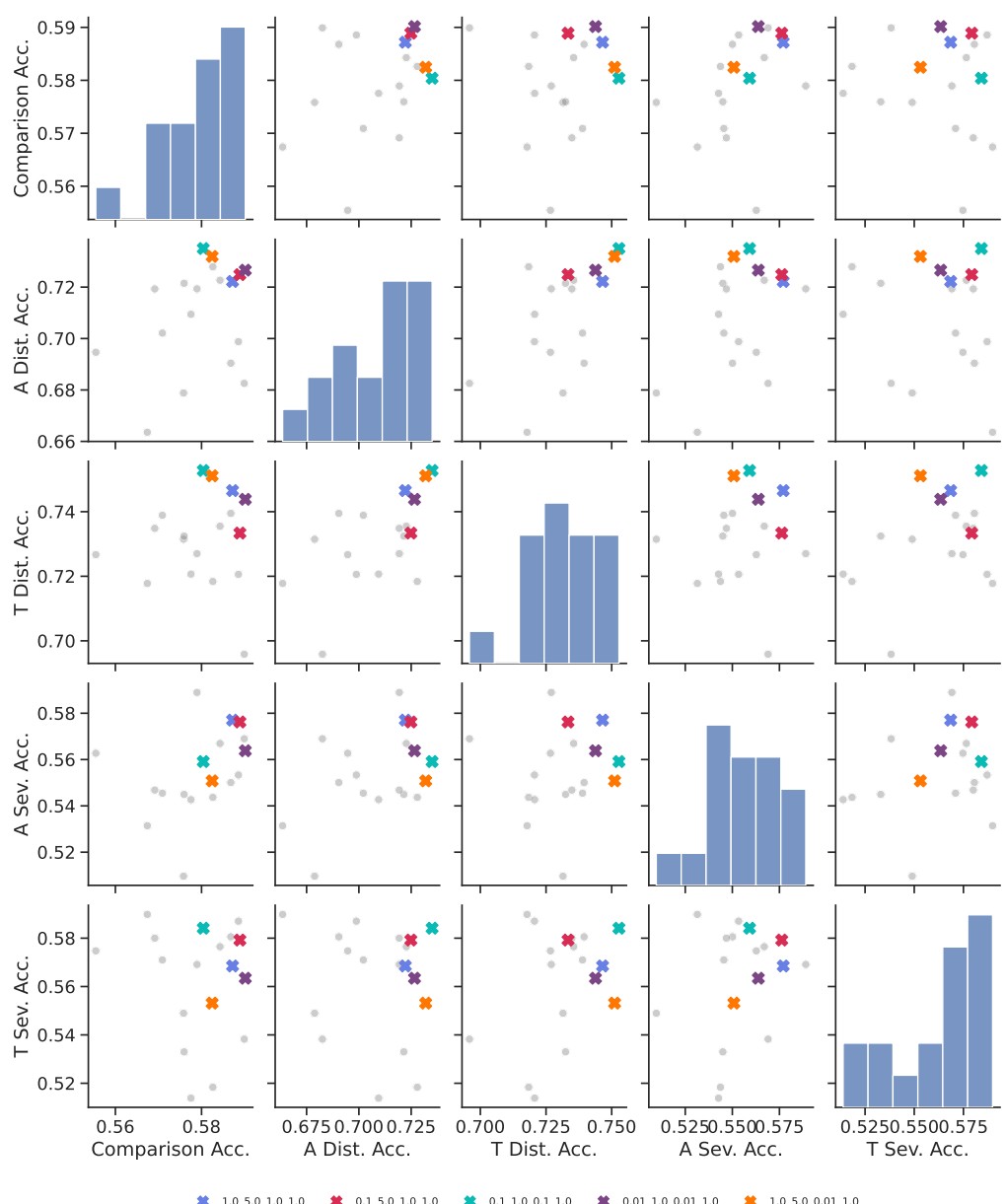

Figure 16: **Hyperparmeter Sweep.** Plot of optimization objective hyperparameter sweep with cross-validation on validation set of PANDASET. Each grey point denotes an experiment that performed noticeably worse, and we label top five settings with colored × mark. A denotes anchor, T denotes Target, Dist. denotes distortion, Sev. is short for severity, and Acc. denotes accuracy.

We, therefore, view PANDASET as the first dataset that enables large-scale, region-wise distortion understanding, and PANDABENCH as the first benchmark that supports systematic evaluation of such region-wise comparative reasoning. We hope that this work catalyzes scientific research on region-grounded comparative quality assessment, and our proposed distortion graph task serves as a foundation towards that end.

## F.2    REPRODUCIBILITY STATEMENT

We provide all the necessary details to reproduce our work, along with architecture details in section 3, experimental setup in section 5, hyperparameter details, and compute requirements in ap-

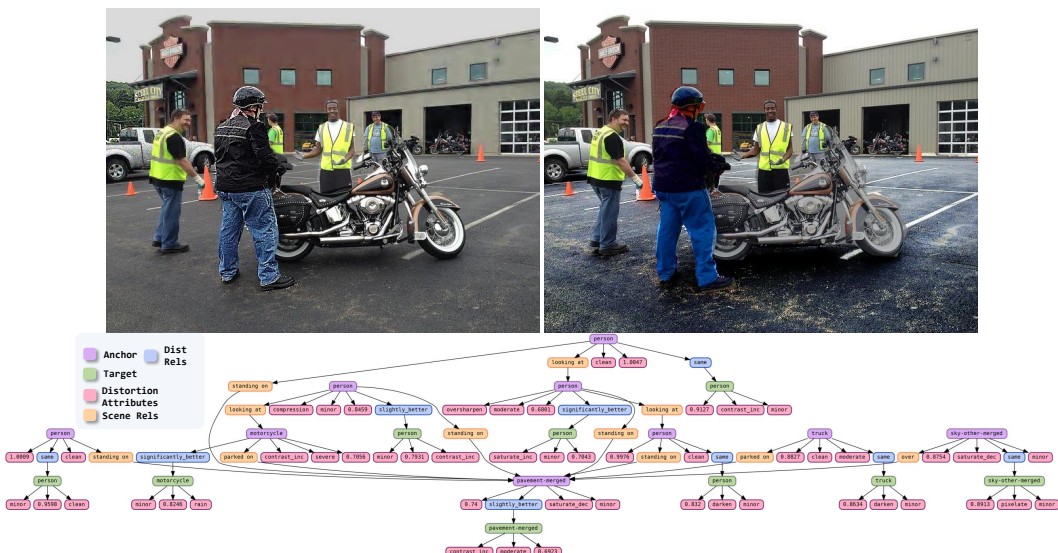

Figure 17: **Dense Distortion Graph Sample.** An example of an image pair with several regions resulting in a dense distortion graph. Left image is Anchor (purple nodes in graph), Right image is Target (green nodes in graph). Legend is presented, and 'rels' is short for relations.

pendix E. We will publicly release our code, trained models, and proposed dataset and benchmark to help further scientific research on comparative assessment and region-level understanding.

## F.3   DATA STATEMENT

As we discuss in section 4, PANDASET is built with two open-source datasets (i) PSG (Yang et al., 2022) and (ii) Seagull-100w (Chen et al., 2024c). Both of these datasets have region-level segmentation maps, and scene information. In PSG, since it is an intersection of COCO Lin et al. (2014) and Visual Genome (Krishna et al., 2017), scene level relationships (or predicates) are provided. While Seagull-100w provides a short description of each region, we use a scene parser (Wu et al., 2019) to parse region relations from these descriptions. Further, images in PANDASET vary in resolution and orientation, e.g., portrait in fig. 13 and landscape in fig. 14, with a minimum spatial resolution of $640 \times 480$. We will release PANDASET with the same license as the original datasets.

