# OpenReview forum: "Panoptic Pairwise Distortion Graph"
_ICLR.cc/2026/Conference — ICLR 2026 Poster_

### Official Review · Reviewer_3yab · 2025-10-26

**Soundness:** 1
**Presentation:** 3
**Contribution:** 3
**Rating:** 4
**Confidence:** 3

**Summary:**

This paper introduces the new task of Distortion Graph (DG), aimed at region-based pairwise image comparison and assessment. The authors argue that existing methods emphasize global analysis while neglecting region-level understanding, and therefore contribute: (1) a region-level dataset, PANDASET, comprising 528K image pairs and 15 distortion types; (2) a three-difficulty benchmark, PANDABENCH; and (3) the PANDA architecture, which employs a DETR-style decoder to predict inter-region comparative relations, distortion types, severities, and quality scores. Experiments show that PANDA substantially outperforms existing MLLMs on PANDABENCH (Tables 2-4), and the generated DGs can serve as chain-of-thought prompts to improve GPT-5 Mini’s accuracy by about 15% (Fig. 4).

**Strengths:**

The core strengths of this work lie in its clear motivation and the comprehensive construction of an ecosystem for the task. Shifting image quality assessment from a global to a structured, region-level representation is a valuable contribution; Fig. 2 compellingly demonstrates failure cases of current MLLMs in understanding region-level distortions. The paper provides a formal definition of DG, a large-scale dataset (2,200 images), a multi-difficulty benchmark (Easy/Medium/Hard), and an efficient baseline model (0.028B parameters). The experimental design is thorough, including systematic comparisons of open- and closed-source MLLMs, ablation studies, and validation of DG as an effective CoT prompt. PANDA’s SOTA performance across all settings (e.g., distortion classification accuracy of 0.78 under the Easy setting) substantiates the method’s effectiveness.

**Weaknesses:**

The most critical limitation of the paper is a circularity issue arising from the construction of the “ground-truth” labels. While the paper aims to propose a fine-grained assessment paradigm superior to existing evaluators, the two most central label types in PANDASET - region-level quality scores and comparative relations (e.g., “slightly better”) - are entirely derived from the existing TOPIQ model: quality scores directly adopt TOPIQ’s full-reference scores, and comparative relations are obtained by thresholding the differences between TOPIQ scores (p. 7; thresholds ±[0.1, 0.3), etc.). This implies that PANDA is essentially learning to imitate TOPIQ’s outputs rather than human perceptual decisions. The paper offers no human annotations to validate the perceptual validity of these automatically generated labels, which weakens the claim that DG can serve as a new evaluation standard. Moreover, the conclusion that “MLLMs cannot understand region-level degradations” (p. 2) may be overstated: the experiments more clearly show that MLLMs, in zero-shot settings, struggle with this novel out-of-domain task (Tables 2-4 indicate near-random performance across all MLLMs), whereas PANDA is trained specifically on PANDASET. Such a comparison is not entirely fair for assessing the fundamental understanding capabilities of MLLMs.

**Questions:**

1. Given that the ground-truth labels for comparative relations and quality scores are entirely dependent on the TOPIQ model, did the authors conduct any human studies to verify that TOPIQ’s region-level scores and their thresholding indeed align with human fine-grained perceptual preferences? If PANDA is merely imitating TOPIQ, how can we ensure it does not inherit TOPIQ’s perceptual biases?

2. The core comparison mechanism of the PANDA decoder allows region features from one image to query features of the other entire image (Eq. 6). Why choose this “region-to-global” attention rather than a more direct “region-to-region” comparison?

3. Fig. 4 shows that using DG as a CoT prompt improves GPT-5 Mini’s performance, yet its original zero-shot accuracy is very low (e.g., comparative accuracy of only 0.31 under the Easy setting, Table 2). Is this improvement due to emergent reasoning capabilities of the MLLM, or is it simply copying answers from PANDA-generated prompts?

4. All models experience a sharp performance drop on PANDABENCH-Hard (PANDA’s distortion accuracy falls from 0.78 to 0.27, Tables 2-3). Is the main cause of this drop that SAM fails to align regions in images with mixed distortions, or that the PANDA decoder cannot effectively compare two inputs both containing complex mixed distortions?

---

> ### Author Response · Authors · 2025-11-21
> **Response to Reviewer Comments -- 1**
>
> We thank the reviewer for their insightful comments, and for finding our contribution valuable, experimental design and evaluation thorough, and motivation clear.
>
> **W1.** Our primary goal in this work is to introduce a representation, Distortion Graph (DG), for comparative image assessment by considering an image pair as a structured composition of its regions. To establish the feasibility of such a representation, we introduced a large-scale dataset and a benchmark as a testbet. PandaBench allows testing region-level comparative reasoning under a well-defined controllable signal. For scalability, we bootstrap labels from TOPIQ [1], which itself has been validated against human Mean Opinion Score (MOS) on several datasets. Our objective in this paper is not to propose a new human-subjective ground-truth standard, we do not claim as such either, but to introduce a structured, region-first perspective on comparative image assessment.
>
> While Panda learns to approximate TOPIQ-induced labels, our DG formalism does not prevent the inclusion of human judgements either in the form of opnion scores as attributes or human preferences as relations in the graph. Further, learning to approximate MOS is well-supported by literature [2, 3] and literature even uses approximations to learned MOS as scores, such as approximating TOPIQ [4].
>
> [2] Li, Weiqi, et al. "Q-insight: Understanding image quality via visual reinforcement learning." arXiv preprint arXiv:2503.22679 (2025).
>
> [3] Wu, Haoning, et al. "Q-align: Teaching lmms for visual scoring via discrete text-defined levels." arXiv preprint arXiv:2312.17090 (2023).
>
> [4] Chen, Zewen, et al. "Seagull: No-reference image quality assessment for regions of interest via vision-language instruction tuning." arXiv preprint arXiv:2411.10161 (2024).
>
> In appendix B.1, we show that DG formalism extends to whole image, i.e., it is possible to rank which image is perceptually superior. We considered the KADID10k dataset where ground-truth labels are derived from subjective annotations done by humans through crowd-sourcing, 30 workers in total [5]. Using either Score (attribute) or Predicate (relationship) from the predicted DG, we find that Panda achieves significant accuracy values 78.83%, and 76.90%, respectively (see Table 7 in appendix). This indicates that Panda indeed approximates the human-judgements through TOPIQ since TOPIQ itself is trained on MOS data. We extend this experiment and consider another famous IQA dataset with human-annotated MOS as the label, TID2013 [6].
>
> We adopt the same experimental setup as KADID10k, query SAM on anchor images to generate region-wise masks, and predict Distortion Graph (DG) for each image pair. We consider both the predicate of the predicted DG and the score attribute and measure ranking accuracy with respect to human-annotated MOS, i.e., which image is better? We present the results here, and have added these results and a brief discussion to appendix B.3 (in purple).
>
> |     Methods    |     Accuracy (%) on TID2013 |
> |:--------------:|:------------------:|
> | mPLUG-Owl2  |  48.5   |
> | LLaVA-1.6 | 57.0 |
> | Q-Instruct    | 55.0 |
> | **Panda (Predicate)**    | **78.4** |
> | **Panda (Score)**    | **77.8** |
>
> *Table 1. Whole Image Ranking Accuracy on TID2013. We report accuracy with both Predicate and Score attribute from predicted DG and compare if they align with human-annotated MOS ground truth when ranking two images.*
>
> [5] Lin, Hanhe, Vlad Hosu, and Dietmar Saupe. "KADID-10k: A large-scale artificially distorted IQA database." 2019 Eleventh International Conference on Quality of Multimedia Experience (QoMEX). IEEE, 2019.
>
> [6] Ponomarenko, Nikolay, et al. "Image database TID2013: Peculiarities, results and perspectives." Signal processing: Image communication 30 (2015): 57-77.
>
> Further, in Tables 2,3,4 (in main paper) we did report results on PandaBench by fine-tuning an open-source MLLM, DepictQA [7] (Vicuna-v1.5-7B as LLM backbone), on PandaSet (our proposed dataset), refer to Sec. 5 for details. We re-produce the results (accuracy on PandaBench Easy/Medium/Hard) here for clarity.
>
> |     Methods    |     Comparison     |     Distortion     |      Severity      |    Scores (PLCC)   |
> |:--------------:|:------------------:|:------------------:|:------------------:|:------------------:|
> | DepictQA (ZS)  | N/A / N/A / N/A    | 0.15 / 0.13 / 0.10 | 0.27 / 0.27 / 0.24 | N/A / N/A / N/A    |
> | DepictQA (SFT) | 0.49 / 0.32 / 0.33 | 0.75 / 0.47 / 0.22 | 0.55 / 0.43 / 0.30 | 0.77 / 0.42 / 0.17 |
> | Panda          | 0.58 / 0.44 / 0.40 | 0.78 / 0.52 / 0.27 | 0.59 / 0.47 / 0.33 | 0.83 / 0.61 / 0.38 |
>
> *Table 2. Accuracy on PandaBench Easy/Medium/Hard comparing fine-tuned MLLM with Panda. ZS refers to Zero-Shot, while SFT refers to Supervised Fine-Tuned on PandaSet. DepictQA uses Vicuna-v1.5-7B as the backbone LLM.*

---

> > ### Author Response · Authors · 2025-11-21
> > **Response to Reviewer Comments -- 2**
> >
> > **W1 (Contd.)** We discuss these results in Sec. 5 (5.1), and in L429-434 we specifically mention the results of DepictQA fine-tuned on PandaSet. We follow the original codebase and the instructions publicly released by authors to train DepictQA. Notice how DepictQA still lags behind Panda, despite being a much larger MLLM (7B). Our results should be interpreted as showing that current training regimes, method design, and even prompting practices are insufficient for robust region-wise distortion reasoning, not that MLLMs are incapable in principle. Therefore, these results complement our argument that approaching comparative assessment in a region-first manner is natural and DG provides a formalism for such.
> >
> > [7] You, Zhiyuan, et al. "Descriptive image quality assessment in the wild." arXiv preprint arXiv:2405.18842 (2024).
> >
> > **Q1.** During initial experiments, we manually verified a few IQA metrics, such as CLIP-IQA [8], TOPIQ-FR [9], and Q-Align [3]. Our goal was to make sure that scores reflected the change in quality of image e.g., an image with a degradation with higher severity should score lower than its lower severity counterpart. Among these, we found TOPIQ-FR to align best, and, hence, we opted for it. Panda's strong performance in experiments in B.1 and B.3 (in appendix) on KADID10k and TID2013 confirm that TOPIQ-based scores and predicates align well with human perferences in terms of perceptual quality. We agree that any method relying on TOPIQ (or any other IQA method) will inherit some of its perceptual biases. While it is possible to use multiple IQA methods and utilize an ensembled approach for scores, we intentionally kept it simple.
> >
> > [8] Wang, Jianyi, Kelvin CK Chan, and Chen Change Loy. "Exploring clip for assessing the look and feel of images." Proceedings of the AAAI conference on artificial intelligence. Vol. 37. No. 2. 2023.
> >
> > [9] Chen, Chaofeng, et al. "Topiq: A top-down approach from semantics to distortions for image quality assessment." IEEE Transactions on Image Processing 33 (2024): 2404-2418.
> >
> > **Q2.** We opted for region-to-global cross-attention design because it allows each region to see the entire target image, which is important for cases where a region's counterpart is ambiguous due to changes in severity or degradation. Region-to-global allows the region to search for its best match in the other image's features and the background context helps with better degradation/severity identification.
> >
> > **Q3.** We query GPT-5 Mini with predicted DG (from Panda) and explicitly ask it to use the provided DG as a hint and if pixels disagree with DG, infer from pixels for that region instead. We inspected such cases and found that GPT-5 Mini sometimes overrides predicted DG outputs when they conflict with its own visual understanding, suggesting it is not blindly copying. Nonetheless, part of the gain does come from leveraging Panda's predicted DG, which is exactly the intended use: DG as a tool that the MLLMs can call. We have added an illustrative figure and a discussion (in purple) comparing both outputs in the paper in appendix B (Figure 10) for Easy and Hard cases, see appendix B.2.
> >
> > **Q4.** For all the experiments in Tables. 2,3,4, we used the segmentation masks provided by PSG and Seagull. Hence, there is no issue of misalignment due to SAM. For KADID10k (in Table 7) and TID2013 (in Table 8) experiment, we query SAM to show that DG aggregates to whole image analysis. In that case, we only query SAM on the anchor images, and utilize the same mask for the target since images are same. Panda's decoder must disentangle several region-wise distortions, with only a few Transformer layers, it is likely that it under-fit the hardest cases. Further, the Hard setting of the benchmark is incredibly difficult even for humans when it comes to identifying severity and distortion in the mixed settings. We also consider 'significantly_better' and 'slightly_better' as two different classes, hence even the comparison accuracy suffers a drop in performance because it is often ambiguous to differentiate between the two. In general, we find that several distortions perceptually 'look' similar even to human eyes: see haze vs contrast decrease or compression vs clean in Figure 7. Consider Figure 10 (in appendix B) where the target sky region's distortion is predicted as 'brightness', while ground truth is 'clean'.
> >
> > Note that, we intentionally kept Panda simpler since our goal was to present a minimal yet necessary baseline (as we highlight in Limitations section, see F.1 in appendix) for region-wise comparative assessment.

---

> > > ### Author Response · Authors · 2025-11-28
> > >
> > > Dear Reviewer 3yab,
> > >
> > > Thank you very much for your time and insightful comments. We had provided detailed explanations to clarify the points, and questions raised. As the deadline for the discussion period is quickly approaching, we were just wondering if all of your questions have been addressed. We are committed to answering any questions/concerns that you may have.
> > >
> > > Thank you again for your valuable time.

---

### Official Review · Reviewer_uHjF · 2025-10-27

**Soundness:** 4
**Presentation:** 4
**Contribution:** 4
**Rating:** 8
**Confidence:** 5

**Summary:**

This paper presents a novel paradigm for comparing the quality of image pairs, particularly those with structured region compositions. The core contribution is the distortion graph, a structure designed to represent diverse distortion information, including type, severity, comparison, and quality score. To support this, the authors introduce a new dataset, PANDASET, used to train a graph prediction network. The network is designed in a DETR fashion, with strong backbones (DINOv2 and SAM) and separate decoders to predict the distinct components of the distortion graph. This graph is then used as a structured input to LMMs to generate region-specific distortion summaries for the input image pairs. Experiments demonstrate that this approach significantly improves the region-wise distortion understanding of current LMMs.

**Strengths:**

* The paper is well-written, clearly organized, and easy to follow. The concepts are introduced logically.
* The proposed distortion graph is a novel and intuitive method for structuring the complex task of regional quality comparison, moving beyond simple scalar scores or text descriptions.
* The proposed model architecture is well-motivated and thoughtfully designed, intelligently combining powerful pre-trained backbones (DINOv2, SAM) with a DETR-style architecture suited for object-centric prediction. The use of separate decoders for different graph components is a logical design choice.
* The ability to feed the DG outputs directly into an LMM to generate detailed, region-specific summaries is a practical and valuable application, bridging the gap between quantitative metrics and human-understandable feedback.
* The reported experimental results are impressive, showing a significant improvement in detailed distortion understanding compared to baseline LMMs.

**Weaknesses:**

* **Insufficient Discussion of Related Work:**
    The paper omits a discussion of closely related work in image quality grounding [a, b]. These prior works also aim to localize and describe quality issues, sharing a similar motivation. The paper would be strengthened by citing these works and clearly articulating how the proposed distortion graph paradigm differs from and improves upon existing grounding-based IQA methods. This context is crucial for accurately positioning the paper's contribution within the field.

* **Limitations of the PANDASET Dataset:**
    The newly collected PANDASET dataset presents potential limitations.
    1.  **Scale:** At a reported 2,000 images, the dataset is relatively small, which raises concerns about model overfitting both to the semantics and distortion types.
    2.  **Synthetic Distortions:** The dataset exclusively contains synthetic distortions. This reliance on synthetic data creates significant concerns: (1) The model may not generalize well to the diverse, subtle, and complex artifacts found in real-world images (e.g., camera noise, motion blur, compression artifacts from unknown codecs). (2) The model might be overfitting to the specific synthetic distortion types included in the dataset, potentially failing to identify or compare unseen distortion types.
    The authors should discuss these limitations and provide experiments (even on a small scale) to test the model's robustness to real-world or unseen distortions.

* **Fairness of Experimental Comparison:**
    The experimental comparison with general-purpose LMMs may be unfair, potentially exaggerating the proposed method's superiority. The proposed network is specifically fine-tuned on the PANDASET dataset, while the baseline LMMs are evaluated in a zero-shot inference setting without task-specific training. The observed significant improvements might primarily stem from this specialized training on the target data distribution.
    It might be better to include this discrepancy in training protocols and discuss it as a confounding factor in the results analysis.

[a] Q-Ground: Image Quality Grounding with Large Multi-modality Models. Chen et al., ACM MM 2024.
[b] Grounding-IQA: Grounding Multimodal Language Model for Image Quality Assessment. Chen et al., arxiv 2024.

**Questions:**

Please see weakness points above.

---

> ### Author Response · Authors · 2025-11-21
> **Response to Reviewer Comments -- 1**
>
> We thank the reviewer for their very positive assessment of our work, including the clarity of the manuscript, the novelty and intuition of the distortion graph (DG) formulation, the design of the PANDA architecture, and the practical value of DG.
>
> **W1.** We appreciate the reviewer pointing out Q-Ground [1] and Grounding-IQA [2]. We have now added both works to the Related Work (Sec. 2, highlighted in purple) and included them in our benchmark summary (Table 1, also in purple).
>
> We briefly summarize and position them relative to Distortion Graph (DG).
>
> Q-Ground [1] introduces QGround100K, built on top of Q-Instruct, a single-image dataset of image, textual descriptions, and region-level segmentation and trains an MLLM to jointly provide the explanation, and pixel-level distortion masks, for 5 distortion types. Its grounding is thus phrased as mapping quality descriptions onto segmentation masks within one image. Grounding-IQA [2] similarly operates in a single-image setting, defining two subtasks: GIQA-DES which considers quality descriptions with bounding boxes and GIQA-VQA which refers to region-wise quality QA. It introduces a dataset GIQA-160K plus GIQA-Bench to fine-tune and evaluate MLLMs on grounding quality attributes to local regions.
>
> In contrast, Distortion Graph (DG) is explicitly comparative in nature and is graph-structured, i.e., it represents an image pair as a region-aligned graph whose attributes convey distortion type, severity, and quality scores, while edges encode ordered comparative relations (e.g., same/slightly_better, etc.). Rather than treating grounding as localizing free-form descriptions, DG formalizes pairwise region-level quality comparison as a structured prediction problem, and the resulting graph can be used as a compact evaluation target (PandaBench) or can be fed as input to general-purpose MLLMs to, optionally, provide language-native descriptions, refer to Figure 2 in the main paper for illustration.
>
> [1] Chen, Chaofeng, et al. "Q-ground: Image quality grounding with large multi-modality models." Proceedings of the 32nd ACM International Conference on Multimedia. 2024.
>
> [2] Chen, Zheng, et al. "Grounding-iqa: Multimodal language grounding model for image quality assessment." arXiv preprint arXiv:2411.17237 (2024).
>
> **W2.** We discuss both limitations that the reviewer pointed out as follows.
>
> **W2a.** The reported 2,200 base images refer to the number of distinct content scenes drawn from PSG and Seagull-100w. We then systematically generate 528K image pairs, and each image has several regions (mean of 18) forming the PandaSet. From a learning perspective, the model does not see 2,200 samples, but it sees hundreds of thousands of diverse pairwise comparisons across a large space of regions, distortion types, and severities. We explain these details in Sec. 4.1 in the main paper.
>
> **W2b.** PandaSet is built on real images and real region masks from PSG and Seagull-100w. We introduced 14+1 distortion types and region-wise severity levels using carefully parameterized operations, and this is similar in spirit to prior work [5]. In case of Seagull-100w, we actually keep the real ISP distortions when they overlap with our chosen distortion (we describe this in detail in Sec. 4.1 in main paper). So PandaSet contains real-world distortions as well, and is not just entirely synthetic.
>
> More importantly, the goal of PandaSet is to provide a controlled, large-scale testbed for region-wise comparative asessment. This control is necessary because we want to fairly evaluate methods that compare two images of similar semantic content with varying region-wise difficulty. Further, this control is what makes it possible to (i) define ground-truth edge predicates deterministically (same/slightly/significantly better/worse), (b) guarantee that every region pair has well-defined reference, (c) systematically vary difficulty (from Easy to Medium to Hard).
>
> The inclusion of Seagull-100w in PandaSet illustrates that it is possible to incorporate real-world distortions, but the lack of a large-scale annotated real-world IQA dataset in literature in general is the limiting factor.
>
> [5] You, Zhiyuan, et al. "Depicting beyond scores: Advancing image quality assessment through multi-modal language models." European Conference on Computer Vision. Cham: Springer Nature Switzerland, 2024
>
> Following reviewer's suggestion, we provide additional results on external datasets, KADID10k and TID2013. Both are famous IQA datasets with a variety of distortion types, severities, and associated human-annotated MOS labels with each image. We consider the task of ranking whole images, i.e., given two images, we use predicted DG's predicate (comparative relations) and score attribute to compare which image is perceptually better. Strong results on both of these datasets indicate that Panda generalizes and the DG formalism is not dependent on PandaSet. We present these results in Sections B.1 and B.3 in appendix.

---

> ### Author Response · Authors · 2025-11-21
> **Response to Reviewer Comments -- 2**
>
> **W3.** In Tables 2,3,4 (in main paper) we did report results on PandaBench by fine-tuning an open-source MLLM, DepictQA [7] (Vicuna-v1.5-7B as LLM backbone), on PandaSet (our proposed dataset), refer to Sec. 5 for details. We re-produce the results (accuracy on PandaBench Easy/Medium/Hard) here for clarity.
>
> |     Methods    |     Comparison     |     Distortion     |      Severity      |    Scores (PLCC)   |
> |:--------------:|:------------------:|:------------------:|:------------------:|:------------------:|
> | DepictQA (ZS)  | N/A / N/A / N/A    | 0.15 / 0.13 / 0.10 | 0.27 / 0.27 / 0.24 | N/A / N/A / N/A    |
> | DepictQA (SFT) | 0.49 / 0.32 / 0.33 | 0.75 / 0.47 / 0.22 | 0.55 / 0.43 / 0.30 | 0.77 / 0.42 / 0.17 |
> | Panda          | 0.58 / 0.44 / 0.40 | 0.78 / 0.52 / 0.27 | 0.59 / 0.47 / 0.33 | 0.83 / 0.61 / 0.38 |
>
> *Table 1. Accuracy on PandaBench Easy/Medium/Hard comparing fine-tuned MLLM with Panda. ZS refers to Zero-Shot, while SFT refers to Supervised Fine-Tuned on PandaSet. DepictQA uses Vicuna-v1.5-7B as the backbone LLM.*
>
> We discuss these results in Sec. 5 (5.1), and in L429-434 we specifically mention the results of DepictQA fine-tuned on PandaSet. We follow the original codebase and the instructions publicly released by authors to finetune DepictQA. Notice how DepictQA still lags behind Panda, despite being a much larger model (7B). Our results should be interpreted as showing that current training regimes, method design, and even prompting practices are insufficient for robust region-wise distortion reasoning. Therefore, these results complement our argument that approaching comparative assessment in a region-first manner is natural and DG provides a formalism for such.
>
> [7] You, Zhiyuan, et al. "Descriptive image quality assessment in the wild." arXiv preprint arXiv:2405.18842 (2024).

---

### Official Review · Reviewer_w8pd · 2025-10-28

**Soundness:** 3
**Presentation:** 3
**Contribution:** 3
**Rating:** 6
**Confidence:** 5

**Summary:**

This paper introduces Distortion Graphs (DG) that enable regional distortion analysis through topological representations of image regions. The authors propose a region-level training dataset, an evaluation benchmark, and an efficient architecture for generating distortion maps. Extensive experiments demonstrate the effectiveness of Panda in regional distortion analysis.

**Strengths:**

1. Regional distortion evaluation through DG is reasonable and effective, and this is the first explicit pairwise regional distortion assessment.
2. The proposed dataset PandaSet and benchmark PandaBench can promote development in the related research community.
3. Comparisons with existing MLLM approaches, including both open-source and closed-source models, support the effectiveness of the proposed regional distortion analysis.
4. The paper contains rich content, including numerous figures and tables that improve clarity.
5. Supplementary materials provide additional examples and code, which further strengthen the solid.

**Weaknesses:**

1. The discussion on region-aware understanding in IQA is insufficient. Prior works such as [1] and [2] should be included to improve motivation and task background.
2. The evaluation is mainly conducted on PandaBench, which limits the ability to fully assess performance. It is recommended to expand the evaluation to more general datasets or conduct user studies to validate generalization further.



[1] Q-Ground: Image Quality Grounding with Large Multi-modality Models

[2] Grounding-IQA: Grounding Multimodal Language Model for Image Quality Assessment

**Questions:**

1. More analysis and discussion of IQA methods that incorporate regional understanding.
2. Broader evaluation to verify generalization capability.

---

> ### Author Response · Authors · 2025-11-21
> **Response to Reviewer Comments**
>
> We thank the reviewer for their thoughtful comments and insights, and for finding our proposed task, DG, effective, the proposed dataset and benchmark important, and manuscript comprehensive and clear.
>
> **W1/Q1.** We thank the reviewer for pointing out two references. We briefly discuss both here and have added discussion to related work in the main paper, see Sec. 2, and benchmarks to Table 1 (both in purple).
>
> Q-Ground [1] introduces QGround100K, built on top of Q-Instruct, a single-image dataset of image, textual descriptions, and region-level segmentation and trains an MLLM to jointly provide the explanation, and pixel-level distortion masks, for 5 distortion types. Its grounding is thus phrased as mapping quality descriptions onto segmentation masks within one image. Grounding-IQA [2] similarly operates in a single-image setting, defining two subtasks: GIQA-DES which considers quality descriptions with bounding boxes and GIQA-VQA which refers to region-wise quality QA. It introduces a dataset GIQA-160K plus GIQA-Bench to fine-tune and evaluate MLLMs on grounding quality attributes to local regions.
>
> In contrast, Distortion Graph (DG) is explicitly comparative in nature and is graph-structured, i.e., it represents an image pair as a region-aligned graph whose attributes convey distortion type, severity, and quality scores, while edges encode ordered comparative relations (e.g., same/slightly_better, etc.). Rather than treating grounding as localizing free-form descriptions, DG formalizes pairwise region-level quality comparison as a structured prediction problem, and the resulting graph can be used as a compact evaluation target (PandaBench) or can be fed as input to general-purpose MLLMs to, optionally, provide language-native descriptions, refer to Figure 2 in the main paper for illustration.
>
> [1] Chen, Chaofeng, et al. "Q-ground: Image quality grounding with large multi-modality models." Proceedings of the 32nd ACM International Conference on Multimedia. 2024.
>
> [2] Chen, Zheng, et al. "Grounding-iqa: Multimodal language grounding model for image quality assessment." arXiv preprint arXiv:2411.17237 (2024).
>
> **W2/Q2.** In Table 7 (in appendix), we considered whole-image ranking task with KADID10k [3] dataset. Using Panda's DG outputs, we rank images from KADID10k, which is labeled with human Mean Opinion Scores via crowdsourcing. Panda achieves 78.83% and 76.90% accuracy using DG scores and predicates, respectively, indicating that DG formalism can aggregate to whole image assessments, and Panda can transfer to a human-MOS benchmark at the image level.
>
> [3] Lin, Hanhe, Vlad Hosu, and Dietmar Saupe. "KADID-10k: A large-scale artificially distorted IQA database." 2019 Eleventh International Conference on Quality of Multimedia Experience (QoMEX). IEEE, 2019.
>
> As per reviewer's request, we conducted another experiment on TID2013 [4] dataset, which is another popular IQA dataset with 2500 images degraded by several distortions, and human-annotated MOS scores are provided per image. We adopt the same experimental setup as KADID10k, query SAM on anchor images to generate region-wise masks, and predict Distortion Graph (DG) for each image pair. We consider both the predicate of the predicted DG and the score attribute and measure ranking accuracy with respect to human-annotated MOS, i.e., which image is better? We present the results here, and have added these results and a brief discussion to appendix B.3 (in purple).
>
> |     Methods    |     Accuracy (%) on TID2013 |
> |:--------------:|:------------------:|
> | mPLUG-Owl2  |  48.5   |
> | LLaVA-1.6 | 57.0 |
> | Q-Instruct    | 55.0 |
> | **Panda (Predicate)**    | **78.4** |
> | **Panda (Score)**    | **77.8** |
>
> *Table 1. Whole Image Ranking Accuracy on TID2013. We report accuracy with both Predicate and Score attribute from predicted DG and compare if they align with human-annotated MOS ground truth when ranking two images.*
>
> [4] Ponomarenko, Nikolay, et al. "Image database TID2013: Peculiarities, results and perspectives." Signal processing: Image communication 30 (2015): 57-77.
>
> It is possible to do whole image ranking since we query SAM to segment the image into regions and predict region-grounded DG. However, it is important to note that our current evaluation focuses on PandaBench (Tables 2,3,4) largely out of necessity. To the best of our knowledge, there is no existing dataset that is simultaneously (i) pairwise comparative, (ii) region-first, and (iii) provides dense distortion annotations, for diverse set of distortions, at the region level (distortion type, severity, quality scores) plus region-wise comparative labels between two images. This lack of suitable benchmark is precisely why we introduce PandaSet/PandaBench in the first place.

---

> > ### Comment · Reviewer_w8pd · 2025-11-27
> >
> > Thank you for the authors’ response. The response have addressed my concerns.

---

> ### Author Response · Authors · 2025-11-27
>
> Dear Reviewer w8pd,
>
> Thank you for your reply. We appreciate your comments and time spent on reviewing this paper and are glad that our response has addressed your concerns. In the rebuttal, we have addressed your main questions/concerns raised in your review, including:
>
> 1. Related Work: We provided summaries of both Grounding-IQA and Q-Ground and positioned our work, DG, in comparison to these methods. We also expanded our related work section in the main paper to include discussions.
>
> 2. Broader Evaluation: In addition to KADID10k experiments (already in the paper), we also added results on another famous IQA dataset, TID2013. A discussion on these results is also added to the paper in appendix B.3.
>
> We are happy to offer more information if there are any other details that would better represent the contribution of our work. We thank you again for your time and consideration.

---

### Official Review · Reviewer_ZyHe · 2025-11-01

**Soundness:** 3
**Presentation:** 3
**Contribution:** 2
**Rating:** 4
**Confidence:** 4

**Summary:**

This paper introduces a novel task called "distortion graph", which represents regional-level quality comparison between image pairs as a structured graph. In this graph, nodes encode regional attributes such as distortion type, severity, and quality score, while edges represent comparative relationships. To support this task, the authors construct the PandaSet dataset and the PandaBench benchmark, and design an efficient model named Panda. Experiments show that existing multimodal large models perform poorly on regional distortion understanding, while Panda significantly outperforms them.

**Strengths:**

1. It creatively extends the concept of scene graphs to the field of image quality comparison, defining a structured and interpretable representation method.
2. The paper's figures are highly expressive, clearly illustrating both the motivation and performance of the work.
3. The meticulously constructed PandaSet dataset contains over 500,000 image pairs with 15 distortion types and region-level annotations, demonstrating substantial scale.

**Weaknesses:**

1. Although the paper introduces the "Distortion Graph" structure, it essentially extends Scene Graph to image-pair comparison tasks. Meanwhile, region-level understanding has already been explored in recent IQA studies such as grounding IQA.

2. PandaSet builds upon PSG and Seagull-100w without providing genuinely new real-distortion data, relying solely on synthetic augmentations.

3. Comparing the specially-designed Panda model with MLLMs not fine-tuned for regional distortion tasks makes the conclusion "MLLMs perform poorly" appear unfair due to task mismatch.

4. Ablation studies remain insufficient, lacking evidence of how existing open-source vision models (Qwen, LLava, Kimi-vl) would perform after fine-tuning on the authors' dataset.

5. The paper provides limited analysis of the token-pool module, region-alignment mechanism, and loss-weight balancing strategies.

**Questions:**

1. If the visual encoder were replaced by a fine-grained, structure-preserving model like DeepSeek-OCR, would it be capable of capturing the complex scene relationships in the images? We look forward to the authors’ insightful answer.

2. Does the stability of SAM-based segmentation affect the precision of region-level alignment?

3. Is the proposed dataset organized in a manner analogous to the Tree-of-Thoughts (ToT) approach? The authors are requested to provide a comparative analysis, along with an evaluation of the associated training and inference time costs.

4. Is this dataset inherently more suitable for video understanding scenarios by nature?

---

> ### Author Response · Authors · 2025-11-21
> **Response to Reviewer Comments -- 1**
>
> We thank the reviewer for their constructive comments and for finding our work creative, figures highly expressive, and dataset meticulous.
>
> **W1.** Unlike classical scene graphs, which operate within a single image and encode semantic relations (e.g., 'person-driving-car'), Distortion Graph (DG) is defined over pairs of images with a structure specialized for comparative assessment, and incorporates intra-image semantic relations. On its own, scene graph structure is not sufficient for comparative assessment problems since it cannot describe how two visual inputs (e.g., images) related to one another.
>
> Grounding-IQA: Grounding IQA operates in a single-image setting, defining two subtasks: GIQA-DES which considers quality descriptions with bounding boxes and GIQA-VQA which refers to region-wise quality QA. It introduces a dataset GIQA-160K plus GIQA-Bench to fine-tune and evaluate MLLMs on grounding quality attributes to local regions. It is important to note that Grounding-IQA is not a comparative assessment method. In contrast, DG approaches the problem in a comparative manner and defines a joint interpretable structure over both images enabling region-wise comparative assessment. Once region-wise comparative relations and attributes such as degradation type, severity and quality score are learned, doing region-wise degradation QA is trivial. In our work, we opted for Seagull [1] as the region-wise IQA method for comparison/analysis, see Tables 2, 3, 4 in the paper, because Grounding-IQA did not open-source their code or dataset, see [Github](https://github.com/zhengchen1999/Grounding-IQA). We have now added a discussion on Grounding-IQA in the related work section in the manuscript (in color purple).
>
> Notably, we do not argue that this work is the first step towards region-wise quality assessment. We position DG as the first step towards comparative assessment grounded in regions and posit that imposing a structured formalism over the problem of region-wise comparative analysis is a natural way towards the solution.
>
> [1] Chen, Zewen, et al. "Seagull: No-reference image quality assessment for regions of interest via vision-language instruction tuning." arXiv preprint arXiv:2411.10161 (2024).
>
> **W2.** PandaSet is built on real images and real region masks from PSG and Seagull-100w. We introduced 14+1 distortion types and region-wise severity levels using carefully parameterized operations, and this is similar in spirit to prior work [2]. In case of Seagull-100w, we actually keep the real ISP distortions when they overlap with our chosen distortion (we describe this in detail in Sec. 4.1 in main paper), so PandaSet contains real-world distortions as well, and is not just entirely synthetic.
>
> More importantly, the goal of PandaSet is to provide a controlled, large-scale testbed for region-wise comparative asessment. This control is necessary because we want to fairly evaluate methods that compare two images of similar semantic content with varying region-wise difficulty. Further, this control is what makes it possible to (i) define ground-truth edge predicates deterministically (same/slightly/significantly better/worse), (b) guarantee that every region pair has well-defined reference, (c) systematically vary difficulty (from Easy to Medium to Hard).
>
> The inclusion of Seagull-100w in PandaSet illustrates that it is possible to incorporate real-world distortions, but the lack of a large-scale annotated real-world IQA dataset in literature in general is the limiting factor.
>
> [2] You, Zhiyuan, et al. "Descriptive image quality assessment in the wild." arXiv preprint arXiv:2405.18842 (2024).

---

> > ### Author Response · Authors · 2025-11-21
> > **Response to Reviewer Comments -- 2**
> >
> > **W3.** In Tables 2,3,4 in the main paper, we compare Q-Insight, DepictQA, Seagull and Gemma 3 27B in the open-source/open-weight MLLM category. Other than Gemma 3 27B which is a general-purpose MLLM, all other MLLMs are distortion-specific and their training data has a significant overlap in distortion types, severity levels and quality scores with PandaSet. Other than weather-type distortion, all of our distortion types are the same as the ones in DepictQA and have high degree of overlap with Seagull. Q-Insight also uses the degradation dataset introduced in DepictQA. We discuss this at length in Sec. 4 in the main paper.
> >
> > All of these methods are specifically designed for distortion analysis as well. We clearly state in the manuscript that even these distortion-specific MLLMs suffer when reasoning about distortion in terms of regions because they are instruction-tuned (which makes them rigid in instruction following) [3], have limited context length and cannot account for all regions in an image, and often do not utilize their vision encoders effectively [4]. We also acknowledged, in the main paper, that the performance gap in general-purpose open-source/open-weight MLLMs (such as Gemma 3 27B/Janus-Pro-7B) should be attributed to lack of exposure to degraded images (L374-376).
> >
> > Therefore, we position DG, and Panda, such that it can complement MLLMs in offering region-wise distortion analysis in natural language via the MLLMs (L092-093 in the main paper). Additionally, in appendix C we extensively document several open-source MLLMs, both distortion-specific such as Q-Instruct, Co-Instruct and general-purpose such as Janus-Pro-7B, and discuss their behavior.
> >
> > [3] Chu, Tianzhe, et al. "Sft memorizes, rl generalizes: A comparative study of foundation model post-training." arXiv preprint arXiv:2501.17161 (2025).
> >
> > [4] Fu, Stephanie, et al. "Hidden in plain sight: VLMs overlook their visual representations." arXiv preprint arXiv:2506.08008 (2025).
> >
> > We emphasize that our work provides extensive evidence that current training recipes, architecture design, etc. do not equip MLLMs with robust region-wise distortion understanding, and that DG formalism is more appropriate for such comparative problems as it can complement the underlying MLLM, if required.
> >
> > **W4.** We ablated several components of the proposed method, Panda, including the backbone choice, decoder design, and sufficiency of backbone, see appendix B for details. Further, in Tables 2,3,4 we did report results on PandaBench by fine-tuning an open-source MLLM, DepictQA [2] (with Vicuna-v1.5-7B as LLM backbone), on PandaSet (our proposed dataset), refer to Sec. 5 for details. We re-produce the results (accuracy on PandaBench Easy/Medium/Hard) here for clarity.
> >
> > |     Methods    |     Comparison     |     Distortion     |      Severity      |    Scores (PLCC)   |
> > |:--------------:|:------------------:|:------------------:|:------------------:|:------------------:|
> > | DepictQA (ZS)  | N/A / N/A / N/A    | 0.15 / 0.13 / 0.10 | 0.27 / 0.27 / 0.24 | N/A / N/A / N/A    |
> > | DepictQA (SFT) | 0.49 / 0.32 / 0.33 | 0.75 / 0.47 / 0.22 | 0.55 / 0.43 / 0.30 | 0.77 / 0.42 / 0.17 |
> > | Panda          | 0.58 / 0.44 / 0.40 | 0.78 / 0.52 / 0.27 | 0.59 / 0.47 / 0.33 | 0.83 / 0.61 / 0.38 |
> >
> > *Table 1. Accuracy on PandaBench Easy/Medium/Hard comparing fine-tuned MLLM with Panda. ZS refers to Zero-Shot, while SFT refers to Supervised Fine-Tuned on PandaSet. DepictQA uses Vicuna-v1.5-7B as the backbone LLM.*
> >
> > We discuss these results in Sec. 5 (5.1), and in L429-434 we specifically mention the results of DepictQA fine-tuned on PandaSet. Notice how DepictQA still lags behind Panda, despite being a much larger MLLM (7B). These results show that MLLM discussion is not limited to zero-shot scenarios, and even when fine-tuned on PandaSet, a region-first design choice has advantages. These results complement our argument that approaching comparative assessment in a region-first manner is natural and DG provides a formalism for such.

---

> > > ### Author Response · Authors · 2025-11-21
> > > **Response to Reviewer Comments -- 3**
> > >
> > > **W5.** In Sec. 3, we introduce the token pool as a way to associate regions with their underlying image, using learnable spatial tokens that are modulated by masks and fused with encoder features. The important property here is that it allows a variable number of regions per image without incurring a significant computational cost.
> > >
> > > Region alignment in PandaSet comes from the shared panoptic segmentation across pairs, i.e., distortions are applied to both anchor and target using same region masks. So for training/evaluation on PandaSet, alignment is exactly inherited from PSG/Seagull segmentations and is not sensitive to SAM, see Sec. 4 (L286-292). SAM is used when we transfer to unseen datasets such as KADID10k and TID2013 to show that DG formalism generalizes to whole image comparisons and human-annotated MOS benchmarks, see appendix B.1 and B.3. In this case, we make sure to pick regions only from the anchor, since anchor and target are similar in content, therefore, no issue of alignment arises. We kept this explicitly simple to avoid introducing additional components in the method.
> > >
> > > We also did extensive hyperparameter analysis, see appendix E. We explain that we sweep through each hyperparameter $\lambda \in \{0.01, 0.1, 1.0, 5.0\}$ and learning rate $\in \{1e^{-2}, 1e^{-4}, 1e^{-6}\}$ using cross-validation. The results of the sweeps are visualized in Figure 15 in the appendix. We observed that Panda is not overly sensitive to hyperparameter selection, and that reasonable choices are sufficient for stable performance (L1130-L1133).
> > >
> > > **Q1.** Panda, our method, is encoder agnostic by design. We ablate different encoder choices (see Figure 6 in appendix). Panda treats encoder as a drop-in module, and any backbone that outputs a spatial feature map can be used in principle. Our token-pool and degradation decoder would remain unchanged. However, distortions primarily affect low-level statistics, for which DINOv2 type encoders are more appropriate. DeepSeek-OCR adopts SAM for local attention and CLIP-Large for global attention along with compression techniques. In our ablations, we experimented with SigLip as the backbone, which is a CLIP-style encoder. We found that DINOv2 worked the best in extracting necessary low-level details.
> > >
> > > **Q2.** The task of comparative image assessment requires that the two regions that are to be compared are similar in content (or have a high-degree of overlap). Therefore, in experiments where segmentation masks are not available, e.g., KADID10k and TID2013 experiments in appendix B.1 and B.3, respectively, we make sure to only call SAM on anchor images and use same region for corresponding target images. Hence, there is no issue of region-level alignment. We kept this explicitly simple to avoid introducing additional components in the method. Note that, in case of view-point changes across images, it is possible to select masks based on IoU thresholding, i.e., measuring overlap between two regions, or enforcing matching during learning.
> > >
> > > **Q3.** DG is a graph-structured representation over regions, and does not involve an explicit tree search, branching, or iterative thought like ToT. However, DG can be used within a ToT-style reasoning process (as an intermediate stage) if desired. Further, Panda generates DG for a single image with 14 regions in about 3.53 seconds on a single v100 GPU, which is orders of magnitude faster than a setup where an MLLM repeatedly queries regions. We compare the cost of different MLLMs considered in the work, along with Panda, in Table 6 (in appendix) and discuss in appendix B.
> > >
> > > **Q4.** For pairwise assessment of videos, it is indeed possible to extend DG formalism wherein ith frame from video 1 and video 2 serve as anchor and target in the image pair. An edge between two DGs can showcase temporal transition from one frame to another in the video. PandaSet is a comparative image assessment dataset, i.e., it compares two images. While inferencing videos frame-by-frame would extend the dataset to videos, but degradations that are only possible in videos, i.e., temporal jittering, flickering, etc., would require extending the dataset. We leave this extension to future work.

---

> > > > ### Author Response · Authors · 2025-11-28
> > > >
> > > > Dear Reviewer ZyHe,
> > > >
> > > > Thank you very much for your time and insightful comments. We had provided detailed explanations to clarify the points, and questions raised. As the deadline for the discussion period is quickly approaching, we were just wondering if all of your questions have been addressed. We are committed to answering any questions/concerns that you may have.
> > > >
> > > > Thank you again for your valuable time.

---

### Author Response · Authors · 2025-12-01
**Summary of Reviewer Comments and Authors' Responses**

We thank all the reviewers for their time and helpful comments. We are glad that reviewers found our work creative and valuable, figures expressive, dataset meticulous (ZyHe), experiments thorough (3yab), task novel and effective (uHjF, w8pd). We are also thankful to reviewer w8pd for pointing out that our comments addressed their concerns.

**1. PandaSet Contains Synthetic Distortions (ZyHe, uHjF)**

PandaSet is built on real images and real region masks from PSG and Seagull-100w. We introduced various distortion types and region-wise severity levels using carefully parameterized operations. In case of Seagull-100w, we actually keep the real ISP distortions when they overlap with our chosen distortion (we describe this in detail in Sec. 4.1 in main paper). So, PandaSet contains real-world distortions as well, and is not just entirely synthetic.

More importantly, the goal of PandaSet is to provide a controlled, large-scale testbed for region-wise comparative assessment. This control is necessary because we want to fairly evaluate methods that compare two images of similar semantic content with varying region-wise difficulty. Further, this control is what makes it possible to define ground-truth edge predicates (same/slightly/significantly better/worse), guarantee that every region pair has well-defined reference, systematically vary difficulty (from Easy to Medium to Hard). The inclusion of Seagull-100w in PandaSet illustrates that it is possible to incorporate real-world distortions.

**2. Generalization to Other Datasets (w8pd, 3yab)**

In Table 7 (in appendix), we considered whole-image ranking task with KADID10k dataset. Using Panda's DG outputs, we rank images from KADID10k, which is labeled with human-annotated MOS via crowdsourcing. Panda achieves 78.83% and 76.90% accuracy using DG scores and predicates, respectively, indicating that DG formalism can aggregate to whole image assessments, and Panda can generalize and transfer to a human-annotated MOS dataset.

We now conducted experiment on TID2013 dataset, which is another popular IQA dataset with 2500 images degraded by several distortions, and human-annotated MOS scores are provided per image. We adopt the same experimental setup as KADID10k, and predict Distortion Graph (DG) for each image pair. We consider both the predicate of the predicted DG and the score attribute and measure ranking accuracy with respect to human-annotated MOS, i.e., which image is better? We present the results here, and have added these results and a brief discussion to appendix B.3 (in purple).

|     Methods    |     Accuracy (%) on TID2013 |
|:--------------:|:------------------:|
| mPLUG-Owl2  |  48.5   |
| LLaVA-1.6 | 57.0 |
| Q-Instruct    | 55.0 |
| **Panda (Predicate)**    | **78.4** |
| **Panda (Score)**    | **77.8** |

*Table 1. Whole Image Ranking Accuracy on TID2013. We report accuracy with both Predicate and Score attribute from predicted DG and compare if they align with human-annotated MOS ground truth when ranking two images.*

**3. Comparing Panda with MLLM Trained on PandaSet (ZyHe, uHjF, 3yab)**

In Tables 2,3,4 (in main paper) we did report results on PandaBench by fine-tuning an open-source MLLM, DepictQA (Vicuna-v1.5-7B as LLM backbone), on PandaSet (our proposed dataset), refer to Sec. 5 for details. We re-produce the results (accuracy on PandaBench Easy/Medium/Hard) here for clarity.

|     Methods    |     Comparison     |     Distortion     |      Severity      |    Scores (PLCC)   |
|:--------------:|:------------------:|:------------------:|:------------------:|:------------------:|
| DepictQA (ZS)  | N/A / N/A / N/A    | 0.15 / 0.13 / 0.10 | 0.27 / 0.27 / 0.24 | N/A / N/A / N/A    |
| DepictQA (SFT) | 0.49 / 0.32 / 0.33 | 0.75 / 0.47 / 0.22 | 0.55 / 0.43 / 0.30 | 0.77 / 0.42 / 0.17 |
| Panda          | 0.58 / 0.44 / 0.40 | 0.78 / 0.52 / 0.27 | 0.59 / 0.47 / 0.33 | 0.83 / 0.61 / 0.38 |

*Table 1. Accuracy on PandaBench comparing fine-tuned MLLM with Panda. ZS refers to Zero-Shot, while SFT refers to Supervised Fine-Tuned on PandaSet.*

We discuss these results in Sec. 5 (5.1), and in L429-434 we specifically mention the results of DepictQA fine-tuned on PandaSet. We follow the original codebase and the instructions publicly released by authors to finetune DepictQA. Notice how DepictQA still lags behind Panda, despite being a much larger model (7B). Our results should be interpreted as showing that current training regimes, method design, and even prompting practices are insufficient for robust region-wise distortion reasoning. These results complement our argument that approaching comparative assessment in a region-first manner is natural and DG provides a formalism for such.

**4. Limited Related Work (uHjF, w8pd, ZyHe)**

We have added a discussion on both Q-Ground and Grounding-IQA to the main paper in Related Work section (in purple).

---

### Meta-Review · Area_Chair_KA8e · 2026-01-03

**Summary:**

The reviews are mixed, but overall the paper presents a clear and practical contribution: introducing a Distortion Graph (DG) formalism for region-first pairwise distortion comparison, along with PandaSet/PandaBench and an efficient Panda model. Reviewers generally find the motivation strong and the proposed formulation useful for structured region-wise assessment, with strong empirical gains over existing MLLMs in the benchmark settings.

The main concerns raised in the reviews were: 1) positioning/novelty relative to existing “quality grounding” works; 2) reliance on primarily synthetic distortions and questions about generalization; 3) fairness of comparing Panda (trained) to general-purpose MLLMs (often zero-shot); 4) potential circularity because key “ground-truth” labels (region scores and comparative relations) are derived from TOPIQ rather than human annotations.

The rebuttal meaningfully addressed several actionable points by: adding missing related work discussions, adding broader evaluation, clarifying that PandaSet includes cases where real ISP distortions are retained/overlap, and adding a fairer baseline via supervised fine-tuning of DepictQA on PandaSet. One reviewer explicitly confirmed the rebuttal addressed their concerns. Therefore, I lean toward accepting this work.

**Reviewer Concerns:**

The main concerns raised in the reviews were: 1) positioning/novelty relative to existing “quality grounding” works; 2) reliance on primarily synthetic distortions and questions about generalization; 3) fairness of comparing Panda (trained) to general-purpose MLLMs (often zero-shot); 4) potential circularity because key “ground-truth” labels (region scores and comparative relations) are derived from TOPIQ rather than human annotations.

**Reviewer Scores:**

Had the reviewer been able to participate fully in the discussion, I believe their score would likely have remained similar or increased slightly. The discussion helped clarify the contributions, address minor concerns, and align the evaluation across reviewers, which supported the final decision.

---

### Decision · Program_Chairs · 2026-01-26

Accept (Poster)